# PRIVATE TOP-$k$ SELECTION UNDER GUMBEL DIFFERENTIAL PRIVACY GUARANTEES

## ABSTRACT

From the perspective of hypothesis testing, $f$-differential privacy ($f$-DP) as a relaxation of differential privacy (DP) possesses numerous desirable properties, the most prominent of which is its lossless characterization of the composition of DP mechanisms. Within the $f$-DP class, Gaussian differential privacy (GDP), as a canonical family introduced to design Gaussian mechanism, has gained widespread acceptance. However, Gaussian mechanism is not the optimal option for all scenarios to ensure DP. As a type of extreme value distribution, Gumbel distribution is naturally considered to design private top-$k$ selection algorithms. In this work, a new family in $f$-DPs, named Gumbel differential privacy (GumDP), is developed to parameterize Gumbel mechanism as similar to GDP. And the composition of Gumbel mechanisms is studied. In addition, two important composition properties of the Gumbel mechanism are discovered among different private selection problems. Utilizing these, a novel privacy-preserving top-$k$ selection algorithm with Gumbel mechanism, called the peeling algorithm under oneshot RNM, is presented based on the Report Noisy Min (RNM) and peeling algorithms. Simulations demonstrate that the privacy-utility performance of the proposed private selection algorithm is significantly improved compared to the peeling algorithm under RNM with Laplace or Gaussian mechanism.

## 1 INTRODUCTION

With the rapid advancement of the information era, vast amounts of data are generated and released daily. This has led to heightened awareness of personal privacy and increased focus on privacy protection technologies. Based on these, differential privacy (DP) (Dwork et al., 2006a;b), as an emerging technology for protecting individual user privacy, has received widespread attention from both academia and industry. On the one hand, the definition is used by academics for a wide range of research, e.g., the privatization of deep learning (Abadi et al., 2016; Zhao et al., 2020) and federated learning (Wei et al., 2020; Yazdinejad et al., 2024; Cai et al., 2024), and the protection of models and data in statistics (Alparslan & Yıldırım, 2022; Awan & Wang, 2024; Lin et al., 2024; Acharya et al., 2024). On the other hand, in industry, DP is also the core technology used by Apple (Differential Privacy Team, 2017), Google (Erlingsson et al., 2014), Microsoft (Ding et al., 2017), and the US Census Bureau (Abowd, 2018; Groshen & Goroff, 2022).

Under the theoretical framework of DP, designing the privacy-preserving mechanism to perturb the output by adding noise is the core concept of the DP application where the three major ones are Laplace, Gaussian and exponential mechanisms (Dwork et al., 2006a;b; McSherry & Talwar, 2007). With the goal of privacy and utility maximization, a large body of literature examines and parameterizes these mechanisms. However, as stated in (Brenner & Nissim, 2010), there is no universally optimal DP mechanism for all types of queries. Hence, the design of DP mechanism is one of the hot issues in DP research trying to start from the perspective of different noise distributions (Liu, 2019; Sadeghi & Korki, 2022; Muthukrishnan & Kalyani, 2023). In addition, along with the complexity and modularity of the algorithm in large models, there will be multiple queries to the database implying the composition of DP mechanisms. The composition of these mechanisms will degrade the privacy-utility performance. The naive and advanced composition theorems (Dwork et al., 2006a; 2010) are originally formulated to track these privacy performances which only carve out loose privacy upper bounds. To achieve a tighter privacy upper bound, some important variants of DP that have also been proposed to minimize the privacy loss of the composition process are

zero-Concentrated Differential Privacy, Rényi Differential Privacy, truncated Concentrated Differential Privacy (Bun & Steinke, 2016; Mironov, 2017; Bun et al., 2018). According to the above approaches, the composition of Laplace, and Gaussian mechanisms all lead to tighter bounds. Unfortunately, these results are still the relaxing ones of the privacy upper bound. Meanwhile, the statistical perspective of transforming DP into a hypothesis testing problem that cannot distinguish the output of two neighboring datasets under the same mechanism has been proposed to enrich the research perspective (Wasserman & Zhou, 2010). The $f$-differential privacy ($f$-DP) is put forward in line with this idea (Dong et al., 2022) where Gaussian differential privacy (GDP) as a family of $f$-DPs gives a loss-free privacy upper bound carving on the composition of Gaussian mechanisms. And GDP is heavily used because of its lossless after composition and the universality of the Gaussian mechanism (Zheng et al., 2021; Bu et al., 2023; Liu et al., 2024). However, Brenner & Nissim (2010) shows that the Gaussian mechanism is not the optimal one in all application scenarios.

Naturally, the Gumbel distribution, as the most common extreme value distribution, has gained interest in the design of DP mechanisms to protect privacy in selection problems. Among selection problems, the top-$k$ query techniques (Ilyas et al., 2008) is one of the top-mentioned techniques, which are widely used in the Web, medical, government, such as information crawling for search engines, sorting queries for medical data, analyzing and researching demographic data. Obviously, attackers can strike this query process to steal the privacy of individual users. Therefore, this work attempts using Gumbel distribution to privatize the top-$k$ selection algorithm and providing the privacy bound for the $k$-fold Gumbel mechanism embedded implicit in the algorithm. Unfortunately, under the definition of $f$-DP, the lack of research on other types of trade-off functions allows for a tighter privacy characterization for DP mechanisms beyond the Gaussian mechanism. Building upon that, a new family of trade-off functions for Gumbel mechanism is proposed.

## 1.1 RELATED WORKS

The private top-$k$ selection algorithm under DP has been extensively studied in fields such as statistics (Dwork et al., 2021; Cai et al., 2021; Xia & Cai, 2023) and machine learning (Cohen & Lyu, 2023; Lebeda & Tetek, 2025; Pagh et al., 2025). However, how to select a suitable DP mechanism and depict the composition of those mechanisms are still two central issues in designing a top-$k$ selection algorithm under DP. For simplicity, private top-1 query, also called private selection algorithm, is prior subjected to research that returns the minimum perturbed query value $\tilde{q}_i$ and its corresponding index $i$ given $n$ queries $\{q_1, \ldots, q_n\}$. Exponential mechanism (EM) (McSherry & Talwar, 2007; McKenna & Sheldon, 2020) and Report Noisy Min (RNM) algorithm (Dwork & Roth, 2013; Durfee & Rogers, 2019; Zhu & Wang, 2022) are two common selection algorithms under DP. In the development of RNM algorithm, it is essentially a matter of perturbing the output index by attempting to apply the Laplace, Gaussian or Gumbel mechanisms to the query value and re-perturbing the corresponding query value by the Laplace or Gaussian mechanism. Furthermore, extending top-1 to top-$k$, the goal of private top-$k$ selection is to design a DP algorithm that outputs the $k$ smallest perturbed query values $\{\tilde{q}_{i_1}, \ldots, \tilde{q}_{i_k}\}$ and their corresponding indexes $\{i_1, \ldots, i_k\}$. There are two main methods to perform $k$-term items selection, namely peeling algorithm (Hardt & Roth, 2013; Dwork et al., 2021; Xia & Cai, 2023) and oneshot algorithm (Durfee & Rogers, 2019; Qiao et al., 2021). Considering the complexity of analyzing privacy parameters in the oneshot algorithm, this paper considers only the peeling algorithm.

## 1.2 CONTRIBUTIONS

In this work, we try to design a new private top-$k$ selection algorithm with Gumbel mechanism and utilize $f$-DP to ensure better privacy-utility performance for this algorithm. The main contributions of this work are summarized as follows:

- The Gumbel mechanism is firstly proposed to directly noise the query value to ensure DP. Gumbel differential privacy (GumDP), as a special family of trade-off function in $f$-DPs, is designed to precisely characterize the Gumbel mechanism and its composition under the assumption that the query functions are consistent. In addition, two equivalent conversion forms between GumDP and DP are given.

- Two attractive composition properties of the Gumbel distribution in the private selection problem are presented, as seen in Lemma 1 and Lemma 2. Based on these and the RNM

algorithm, a newly validated private selection algorithm with Gumbel mechanism, named oneshot RNM algorithm, is introduced which can simultaneously output the index and query value without re-adding noise. Building upon Gumbel mechanism, Theorem 3 guarantees the privacy of this algorithm.

- Extending to top-$k$ private selection, the peeling algorithm under oneshot RNM with Gumbel mechanism, whose privacy is secured by Theorem 4, is put through the peeling algorithm. And simulations show that there is a significant reduction in the variance of the added noise compared to the peeling algorithm under RNM with Laplace or Gaussian mechanism, which also confirms the increase in data availability and ensures that Gumbel mechanism achieves superior privacy-utility performance in the top-$k$ selection problem.

*Notations:* Let $\mathcal{N}(0, \sigma^2)$, $\mathrm{Lap}(\lambda)$ and $\chi^2(2k)$ represent Gaussian distribution with location parameter 0 and scale parameter $\sigma$, Laplace distribution with mean 0 and scale parameter $\lambda$, and chi-square distribution with parameter $2k$ respectively. The $\mathrm{sign}(x)$ denotes the signature function, i.e., $\mathrm{sign}(x) = \begin{cases} -1, & \text{if } x < 0, \\ 0, & \text{if } x = 0, \\ 1, & \text{if } x > 1. \end{cases}$ And $[m]$ and $\mathcal{R}$ represent the set of $\{1, \cdot, m\}$ and the set of real numbers respectively.

*Mathematical details:* Due to the space limitation, all details of the proofs of lemmas, corollaries and theorems in this paper are provided in the appendices.

## 2 GUMBEL DIFFERENTIAL PRIVACY

For the sake of subsequent discussion, we foresee the definitions of DP and $f$-DP, along with their equivalence transformation relationship. Let $\mathcal{D} = (d_1, d_2, \ldots, d_l)$, $\mathcal{D}' = (d_1', d_2', \ldots, d_l')$, denoted two neighboring datasets containing $l$ data items, of which $l$ can be interpreted as the number of users in the database, be sampling from $\mathcal{X}^l$ where $\mathcal{X}$ is a sample universe. These two datasets differ in one and only one data item, i.e., only one $j \in [l]$ such that $d_j \neq d_j'$. Dwork et al. (2006a;b) propose DP to protect the individual privacy which is unable to distinguish between $\mathcal{D}$ and $\mathcal{D}'$.

**Definition 1** (($\varepsilon, \delta$)-DP (Dwork & Roth, 2013)). *For any $\varepsilon > 0$ and $\delta > 0$, a mechanism $\mathcal{M}$ is $(\varepsilon, \delta)$-DP if for all adjacent databases $\mathcal{D}, \mathcal{D}'$ and any measurable event $S \subset \mathcal{R}$,*

$$P(\mathcal{M}(\mathcal{D}) \in S) \leq \mathrm{e}^\varepsilon P(\mathcal{M}(\mathcal{D}') \in S) + \delta.$$

From the definition of DP, it is evident that the smaller the privacy parameters $\varepsilon$ and $\delta$, the higher the level of privacy protection provided by the corresponding DP mechanism $\mathcal{M}$. In $f$-DP, it is natural to extend it to the problem of hypothesis testing where the distribution of the null hypothesis follows $\mathcal{M}(\mathcal{D})$ and the alternative one follows $\mathcal{M}(\mathcal{D}')$ making it is difficult to distinguish them. Let $\phi$ denote the rejection rule. The trade-off function as a tool to characterize the degree of difference between two hypotheses is

$$T(\mathcal{M}(\mathcal{D}), \mathcal{M}(\mathcal{D}'))(\alpha) = \inf_\phi \{\beta_\phi : \alpha_\phi \leq \alpha\},$$

where $\alpha_\phi$ and $\beta_\phi$ are its corresponding type I and II errors respectively defined as

$$\alpha_\phi = \mathbb{E}_{\mathcal{M}(\mathcal{D})}[\phi], \quad \beta_\phi = 1 - \mathbb{E}_{\mathcal{M}(\mathcal{D}')}[\phi].$$

**Definition 2** ($f$-DP(Dong et al., 2022)). *Given a trade-off function $f : [0, 1] \rightarrow [0, 1]$ satisfies convexity, continuity, and $f(x) \leq 1 - x$ for $x \in [0, 1]$. A mechanism $\mathcal{M}$ is said to be $f$-DP if*

$$T(\mathcal{M}(\mathcal{D}), \mathcal{M}(\mathcal{D}')) \geq f,$$

*for all neighbouring datasets $\mathcal{D}$ and $\mathcal{D}'$.*

The closer $f$ in Definition 2 approaches $g(x) = 1 - x$ with $x \in [0, 1]$, the higher the level of privacy protection provided by the DP mechanism $\mathcal{M}$. Besides, the equivalent conversion between $f$-DP and DP is also given by Dong et al. (2022) through the concept of convex conjugate. For a function $f$ with $f(x) = \infty$ for $x < 0$ or $x > 1$, its convex conjugate is defined as $f^*(y) = \sup_{-\infty < x < \infty}(yx - f(x))$. For a symmetric trade-off function $f$, a mechanism is $f$-DP if and only if it is $(\varepsilon, \delta(\varepsilon))$-DP for all $\varepsilon > 0$ with

$$\delta(\varepsilon) = 1 + f^*(-e^\varepsilon). \tag{1}$$

## 2.1 $\mu$-GUMBEL DIFFERENTIAL PRIVACY

For a random variable $X$ distributed from the Gumbel (minimum) distribution with location parameter $\mu$ and scale parameter $\gamma > 0$, denoted as $X \sim \text{Gum}(\mu, \gamma)$, its variance is $\pi^2 \gamma^2 / 6$, and its cumulative distribution function (CDF) and probability density function (PDF) respectively are

$$F(x; \mu, \gamma) = 1 - e^{-e^{\frac{x-\mu}{\gamma}}}, \; p(x; \mu, \gamma) = \frac{1}{\gamma} e^{\frac{x-\mu}{\gamma} - e^{\frac{x-\mu}{\gamma}}}.$$

**Definition 3** (Gumbel Mechanism). *Given a database $\mathcal{D}$ and a query function $h$, the Gumbel mechanism $\mathcal{M}_{\text{Gum}}$ is defined as*

$$\mathcal{M}_{\text{Gum}}(\mathcal{D}) = h(\mathcal{D}) + \eta, \qquad \eta \sim \text{Gum}(0, \gamma).$$

Analogously to GDP (Dong et al., 2022), from the Gumbel distribution aspect, we design the $\mu$-GumDP as a special family of the trade-off function in $f$-DP. Consider the following hypothesis testing problem:

$$H_0 : y \sim \mathcal{M}_{\text{Gum}}(\mathcal{D}) \quad \text{versus} \quad H_1 : y \sim \mathcal{M}_{\text{Gum}}(\mathcal{D}'). \tag{2}$$

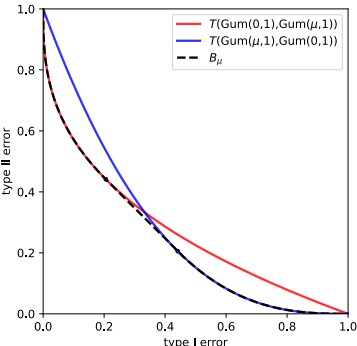

Figure 1: The trade-off functions $T(\text{Gum}(0, 1), \text{Gum}(\mu, 1)), T(\text{Gum}(\mu, 1), \text{Gum}(0, 1))$ and $B_\mu$ with $\mu = 1$.

Unlike the Gaussian distribution, the Gumbel distribution is asymmetric. From the hypothesis testing problem in (2), as shown in the Fig. 1, for any $\mu \geq 0$,

$$T(\text{Gum}(0, 1), \text{Gum}(\mu, 1)) \neq T(\text{Gum}(\mu, 1), \text{Gum}(0, 1)).$$

To facilitate subsequent conversion to DP, we perform a two-step operation in the definition of trade-off function $B_\mu$: symmetrization and convexification. Symmetrization is taking the minimum of $T(\text{Gum}(0, 1), \text{Gum}(\mu, 1))$ and $T(\text{Gum}(\mu, 1), \text{Gum}(0, 1))$; and convexification is taking the bi-conjugate one, i.e., the largest convex lower envelope, of

$$\min\{T(\text{Gum}(0, 1), \text{Gum}(\mu, 1)), T(\text{Gum}(\mu, 1), \text{Gum}(0, 1))\}.$$

**Definition 4.** *For $\mu \geq 0$, the trade-off function $B_\mu$ is defined as*

$$B_\mu = \min\{T(\text{Gum}(0, 1), \text{Gum}(\mu, 1)), T(\text{Gum}(\mu, 1), \text{Gum}(0, 1))\}^{**}. \tag{3}$$

The $B_\mu$ for any $\mu \geq 0$ satisfies the requirements for the trade-off function as defined in Definition 2. Fig. 1 also presents the curve of $B_\mu$ with $\mu = 1$. And the explicit expression for the trade-off function $B_\mu$ in (3) reads

$$B_\mu(\alpha) = \begin{cases} 1 - \alpha^{e^{-\mu}}, & \alpha \in [0, \alpha_1), \\ -\alpha + e^{\frac{\mu}{e^{-\mu}-1}} + 1 - e^{\frac{\mu e^{-\mu}}{e^{-\mu}-1}}, & \alpha \in [\alpha_1, \alpha_2), \\ (1 - \alpha)^{e^\mu}, & \alpha \in [\alpha_2, 1], \end{cases} \tag{4}$$

where $\alpha_1 = e^{\frac{\mu}{e^{-\mu}-1}}$ and $\alpha_2 = 1 - e^{\frac{\mu e^{-\mu}}{e^{-\mu}-1}}$. The proof details of (4) are provided in Appendix A. From (4), this trade-off function is decreasing in $\mu$ that $B_\mu > B_{\mu_0}$ if $\mu < \mu_0$, as shown in Fig. 2(a).

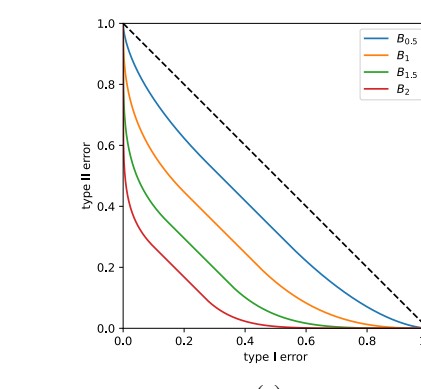 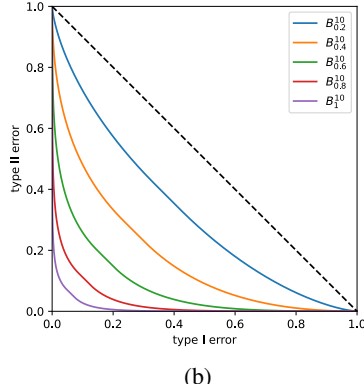

(a)              (b)

Figure 2: (a) Changes in $B_\mu$ curves for different values of $\mu$. (b) Changes in $B_\mu^{10}$ curves for different values of $\mu$.

**Definition 5** (Gumbel Differential Privacy). *A mechanism $\mathcal{M}$ is said to satisfy $\mu$-Gumbel differential privacy, denoted as $\mu$-GumDP, if*

$$T(\mathcal{M}(\mathcal{D}), \mathcal{M}(\mathcal{D}')) \geq B_\mu,$$

*for all neighboring datasets $\mathcal{D}$ and $\mathcal{D}'$.*

**Theorem 1.** *If a Gumbel mechanism operates on a real statistic $h$ as $\mathcal{M}_{\mathrm{Gum}}(\mathcal{D}) = h(\mathcal{D}) + \eta$, where $\eta \sim \mathrm{Gum}(0, \gamma)$ and $\Delta h = \max_{\mathcal{D}, \mathcal{D}'} |h(\mathcal{D}) - h(\mathcal{D}')|$, then $\mathcal{M}_{\mathrm{Gum}}$ is $\mu$-GumDP with $\gamma\mu \geq \Delta h$.*

From Definition 5, $\mu$-GumDP, on the one hand, facilitates the privacy analysis and comparison as a one-parameter privacy definition; on the other hand, achieves a good degree of privacy at $\mu < 0.5$ as shown in Fig. 2(a). Besides, it has a tight privacy carving for the Gumbel mechanism by Theorem 1. Next, we provide an equivalent transformation between $\mu$-GumDP and $(\varepsilon, \delta)$-DP to conveniently compare the privacy-utility performance of different DP mechanisms.

**Corollary 1.** *A mechanism is satisfied $\mu$-GumDP if and only if it is $(\varepsilon, \delta(\varepsilon))$-DP for all $\varepsilon \geq 0$, where $\delta(\varepsilon) = (e^{\mu+\varepsilon} - e^\varepsilon)e^{\frac{\mu+\varepsilon}{e^{-\mu}-1}}$.*

## 2.2 $(k, \mu)$-GUMBEL DIFFERENTIAL PRIVACY

In practical applications, the composition of DP mechanisms is often involved. Therefore, this section proposes $(k, \mu)$-GumDP to characterize the composition of Gumbel mechanisms.

**Definition 6** ($k$-fold Composed Mechanism). *When $k = 2$, with the first mechanism $\mathcal{M}_1 : \mathcal{X}^l \to \mathcal{R}$ and the second mechanism $\mathcal{M}_2 : \mathcal{X}^l \times \mathcal{R} \to \mathcal{R}$, the 2-fold mechanism $\mathcal{M} : \mathcal{X}^l \to \mathcal{R} \times \mathcal{R}$ is given by $\mathcal{M}(\mathcal{D}) = (y_1, \mathcal{M}_2(\mathcal{D}, y_1))$ with $\mathcal{M}_1(\mathcal{D}) = y_1$ and $\mathcal{D} \in \mathcal{X}^l$. Let $\mathcal{M}_i : \mathcal{X}^l \times \mathcal{R}^{i-1} \to \mathcal{R}$, $i \in [k]$. Extension to the case of $k \geq 2$, the $k$-fold composed mechanism $\mathcal{M}$ of $\mathcal{M}_i$, $i \in [k]$, is defined as*

$$\mathcal{M} = (\mathcal{M}_1, \mathcal{M}_2, \ldots, \mathcal{M}_k) : \mathcal{X}^l \to \mathcal{R}^k.$$

Based on Definition 6, considering a new hypothesis testing problem for the $k$-fold composed mechanism $\mathcal{M}$:

$$H_0 : (y_1, y_2, \ldots, y_k) \sim \mathcal{M}(\mathcal{D}) \quad \text{versus} \quad H_1 : (y_1, y_2, \ldots, y_k) \sim \mathcal{M}(\mathcal{D}'). \tag{5}$$

Assuming that $\mathcal{M}_i$ is an independent Gumbel mechanism given $\{y_j\}_{j=1}^{i-1}$, the above hypothesis test can be viewed as a discussion of independent composition of $k$ Gumbel mechanisms. For ease of analysis, the Gumbel mechanism corresponding to each $\mathcal{M}_i$ is based on the identical Gumbel distribution, denoted as $\mathcal{M}_{\mathrm{Gum}}$. Under the above assumptions, $y_1, y_2, \ldots, y_k$ are independently and identically distributed (i.i.d.) from $\mathcal{M}_{\mathrm{Gum}}(\mathcal{D})$ given database $\mathcal{D}$. Then, the hypothesis test problem (5) can be converted to

$$H_0 : \{y_j\}_{j=1}^k \overset{\text{i.i.d.}}{\sim} \mathcal{M}_{\mathrm{Gum}}(\mathcal{D}) \quad \text{versus} \quad H_1 : \{y_j\}_{j=1}^k \overset{\text{i.i.d.}}{\sim} \mathcal{M}_{\mathrm{Gum}}(\mathcal{D}'). \tag{6}$$

Similar to $B_\mu$ in (3), we propose the following trade-off function $B_\mu^k$.

**Definition 7.** *For $\mu \geq 0$, the trade-off function is defined as*

$$B_\mu^k = \min\{T(\mathrm{Gum}(0,1)^k, \mathrm{Gum}(\mu,1)^k), T(\mathrm{Gum}(\mu,1)^k, \mathrm{Gum}(0,1)^k)\}^{**}.$$

And the explicit expression for the trade-off function $B_\mu^k$ reads

$$B_\mu^k = \begin{cases} F_Y(F_Y^{-1}(1-\alpha)e^{-\mu}), & \alpha \in [0, \alpha_1), \\ \alpha_1 - \alpha + F_Y(F_Y^{-1}(1-\alpha_1)e^{-\mu}), & \alpha \in [\alpha_1, \alpha_2), \\ 1 - F_Y(F_Y^{-1}(\alpha)e^\mu), & \alpha \in [\alpha_2, 1], \end{cases} \tag{7}$$

where $\alpha_1 = 1 - F_Y\left(\frac{2k\mu}{1-e^{-\mu}}\right), \alpha_2 = F_Y\left(\frac{2k\mu e^{-\mu}}{1-e^{-\mu}}\right)$ and $Y \sim \chi^2(2k)$ with CDF $F_Y$. The proof details of (7) are provided in Appendix D. The trade-off function $B_\mu^k$ is also decreasing in $\mu$ that $B_\mu^k > B_{\mu_0}^k$ if $\mu < \mu_0$, as seen in Fig. 2b.

**Definition 8** (($k, \mu$)-Gumbel Differential Privacy). *A mechanism $\mathcal{M}$ is said to satisfy $(k, \mu)$-Gumbel differential privacy (($k, \mu$)-GumDP) if*

$$T(\mathcal{M}(\mathcal{D}), \mathcal{M}(\mathcal{D}')) \geq B_\mu^k$$

*for all neighbouring data sets $\mathcal{D}$ and $\mathcal{D}'$.*

Let the query functions $\{h_i\}_{i=1}^k$ be consistent before characterizing the $k$-fold composed mechanism under the Gumbel distribution.

**Definition 9.** *The $k$ query functions $\{h_i\}_{i=1}^k$ are consistent if if either $\mathrm{sign}(h_j(D') - h_j(D)) \leq 0$ for all $j = 1, \ldots, k$, or $\mathrm{sign}(h_j(D') - h_j(D)) \geq 0$ for all $j = 1, \ldots, k$.*

**Theorem 2.** *Consider the Gumbel mechanism operating on a statistic $h_i$ as $\mathcal{M}_i(\mathcal{D}) = h_i(\mathcal{D}, y_1, y_2, \ldots, y_{i-1}) + \eta_i$ where $i \in [k]$, $\eta_i \overset{i.i.d.}{\sim} \mathrm{Gum}(0, \Delta/\mu)$, $\Delta = \max_{i \in [k]} \max_{\mathcal{D}, \mathcal{D}'} |h_i(\mathcal{D}) - h_i(\mathcal{D}')|$. If $\{h_i\}_{i=1}^k$ are consistent, then the $k$-fold composed mechanism $\mathcal{M} = (\mathcal{M}_1, \mathcal{M}_2, \ldots, \mathcal{M}_k)$ is $(k, \mu)$-GumDP.*

By Theorem 2, it can be seen as that the composition of $k$ Gumbel mechanisms satisfied $\mu$-GumDP is $(k, \mu)$-GumDP under certain conditions. Meanwhile, the equivalence transformation between $(k, \mu)$-GumDP and $(\varepsilon, \delta)$-DP is proposed as follows.

**Corollary 2.** *A mechanism is $(k, \mu)$-GumDP if and only if its $k$-fold mechanism is $(\varepsilon, \delta_k(\varepsilon))$-DP for all $\varepsilon > 0$, where $\delta_k(\varepsilon) = 1 - e^\varepsilon + e^\varepsilon F_Z\left(\frac{2(k\mu+\varepsilon)}{1-e^{-\mu}}\right) - F_Z\left(\frac{2(k\mu+\varepsilon)e^{-\mu}}{1-e^{-\mu}}\right)$ and $F_Z$ denotes the CDF of distribution $\chi^2(2k)$.*

Note that the $\delta_k()$ in Corollary 2 is strictly derived from the equivalent conversion between $f$-DP and DP, i.e. Equation (1), so that $\delta_k(\varepsilon) \in [0, 1]$ for all $\varepsilon > 0$.

# 3 PRIVATE TOP-$k$ SELECTION UNDER GUMBEL MECHANISM

To ease the study, the top-$k$ selection problem in this paper is to perform $m$ real queries for any database $\mathcal{D}$, i.e., $\{h_1(\mathcal{D}), h_2(\mathcal{D}), \ldots, h_m(\mathcal{D})\}$, sort the $m$ queries, i.e., $h_{i_1}(\mathcal{D}) < h_{i_2}(\mathcal{D}) < \cdots < h_{i_m}(\mathcal{D})$, and finally output the smallest $k$ query values and the corresponding indexes, i.e., $\{(i_1, h_{i_1}(\mathcal{D})), (i_2, h_{i_2}(\mathcal{D})), \ldots, (i_k, h_{i_k}(\mathcal{D}))\}$. The output of indexes and query values suffers from the leakage of individual privacy in $\mathcal{D}$. The peeling algorithm under RNM as a top-$k$ selection algorithm under DP protects both of them (Dwork et al., 2021). In this section, based on the above algorithm and the Gumbel mechanism under GumDP given in the previous section, we design a newly private top-$k$ selection algorithm. Moreover, analyzing from the perspective of adding noise variance, the new algorithm guarantees higher privacy-utility performance.

## 3.1 THE PEELING ALGORITHM UNDER ONESHOT REPORT NOISE MIN

Before designing the private selection algorithm, there are two important composition properties about Gumbel mechanism.

**Lemma 1.** *Let* $\left\{\mathcal{M}_{\mathrm{Gum}}^{(i)}(\mathcal{D}) = h_i(\mathcal{D}) + \eta_i\right\}_{i \in [m]}$ *be $\mu$-GumDP where* $\{\eta_i\}_{i \in [m]} \stackrel{i.i.d.}{\sim} \mathrm{Gum}(0, \frac{\Delta}{\mu})$. *The minimum Gumbel mechanism $\mathcal{M}_{\mathrm{minGum}}^m$ is defined as, for $m \in \mathbb{N}^+$ and any database $\mathcal{D}$,*

$$\mathcal{M}_{\mathrm{minGum}}^m(\mathcal{D}) = \min_{i \in [m]} \left\{\mathcal{M}_{\mathrm{Gum}}^{(i)}(\mathcal{D})\right\}.$$

*The $\mathcal{M}_{\mathrm{minGum}}^m$ can be also seen as a Gumbel mechanism which satisfies $\mu$-GumDP.*

Lemma 1 illustrates that the minimum output among the noisy query values perturbed by Gumbel noises still satisfies GumDP. However, the private selection problem considered in this paper requires not only the minimum query value but also its corresponding index. To find the best index $j \in [m]$ and output the corresponding query value $h_j(\mathcal{D})$, the RNM algorithm (Dwork & Roth, 2013) with Laplace mechanism satisfied $(\varepsilon, 0)$-DP: Add the independent Laplace noise $\omega$ from $\mathrm{Lap}(2\Delta/\varepsilon)$ to each query $\{h_j(\mathcal{D})\}_{j=1}^m$, return the index $j^*$ of the smallest noisy one $\tilde{h}_{j^*}(\mathcal{D}) = h_{j^*}(\mathcal{D}) + \omega$ and draw a fresh noise $\mathrm{Lap}(2\Delta/\varepsilon)$ added to $h_{j^*}(\mathcal{D})$ to output the noise one. It is evident that the original RNM algorithm suffers from a privacy allocation issue concerning both the index and the query value. Meanwhile, EM in Dwork & Roth (2013) as another common privacy selection algorithm only outputs the best index $i \in [m]$. The EM $\mathcal{M}_{\mathrm{E}}$ satisfies $(\varepsilon, 0)$-DP which outputs the index $i$ with probability

$$P\left(\mathcal{M}_{\mathrm{E}}(\mathcal{D}, \{h_j\}_{j \in [m]}, \varepsilon) = i\right) = \frac{e^{-\frac{\varepsilon h_i(\mathcal{D})}{\Delta}}}{\sum_{j \in [m]} e^{-\frac{\varepsilon h_j(\mathcal{D})}{\Delta}}}.$$

Fortunately, Durfee & Rogers (2019) demonstrates that $\mathcal{M}_{\mathrm{E}}\left(\cdot, \{h_j\}_{j \in [m]}, \varepsilon\right)$ is equivalent to the RNM with $\mathrm{Gum}(0, \Delta/\varepsilon)$ when both of them output only the index. Building upon Lemma 1 and the relation the EM and the Gumbel mechanism, Lemma 2 gives a natural way to assign privacy-preserving parameters to the output query value and its corresponding index in the private selection problem utilizing Gumbel mechanism .

**Lemma 2.** *For any database $\mathcal{D}$ and a batch of query values $\{h_j(\mathcal{D})\}_{j \in [m]}$ added independent noise perturbations from $\mathrm{Gum}(0, \frac{\Delta}{\varepsilon})$, output the minimum noise query value and its index concurrently, denoted as $\mathcal{M}_{\mathrm{Gum}}^*(\mathcal{D})$, is equal to the independent composition of the Gumbel mechanism $\mathcal{M}_{\mathrm{minGum}}^m(\mathcal{D})$ satisfied $\varepsilon$-GumDP and the EM $\mathcal{M}_E(\mathcal{D}, \{h_j\}_{j \in [m]}, \varepsilon)$, i.e., for any $S \subset \mathbb{R}$,*

$$P\left(M_{\mathrm{Gum}}^*(D) = (i, h_i(D) + \eta_i) \in [m] \times S\right)$$
$$= P\left(\mathcal{M}_{\mathrm{minGum}}^m(\mathcal{D}) \in S\right) P\left(\mathcal{M}_E(\mathcal{D}, \{h_j\}_{j \in [m]}, \varepsilon) = i\right).$$

Actually, Lemma 2 also gives a composition of the Gumbel mechanism and the EM. Based on this, the oneshot RNM algorithm which outputs the query and its index at the same time is formulated and presented in Algorithm 1.

---

**Algorithm 1** Oneshot Report Noisy Min

---

**Input:** The database $\mathcal{D}$, functions $h_1, \ldots, h_m$ with sensitivity $\Delta$ and scale parameter $\gamma$
**Output:** The index $j^*$ and approximation to $h_{j^*}(\mathcal{D})$
 1: **for** $j = 1$ to $m$ **do**
 2:     Set $\tilde{h}_j = h_j(\mathcal{D}) + Z_j$, where $Z_j$ is independently sampled from $\mathrm{Gum}(0, \gamma)$;
 3: **end for**
 4: Solve $j^* = \arg\min_{j \in [m]} \tilde{h}_j$ and compute $\tilde{h}_{j^*}$.

---

It is evident that Algorithm 1 is more efficient than the RNM algorithm. Theorem 3 below provides the privacy assurance for this algorithm.

**Theorem 3.** *The oneshot RNM algorithm given in Algorithm 1 is $\left(\frac{\Delta}{\gamma} + \varepsilon, \delta(\varepsilon)\right)$-DP where $\delta(\varepsilon) = \left(e^{\frac{\Delta}{\gamma} + \varepsilon} - e^\varepsilon\right) e^{e^{-\frac{\Delta}{\gamma} - 1}}$ for any $\varepsilon > 0$.*

It is natural to design a new private top-$k$ selection algorithm using the peeling algorithm under oneshot RNM proposed and shown in Algorithm 2. This top-$k$ selection algorithm can be seen as the independent composition of $k$ Gumbel mechanisms satisfied $\frac{\Delta}{\gamma}$-GumDP and $k$ EMs satisfied $\left(\frac{\Delta}{\gamma}, 0\right)$-DP.

---

**Algorithm 2** Peeling Algorithm under Oneshot Report Noisy Min

---

**Input:** database $\mathcal{D}$, functions $h_1, \ldots, h_m$ with sensitivity $\Delta$, number of invocations $k$ and scale parameter $\gamma$

1: **for** $j = 1$ to $k$ **do**
2:    Let $\left(i_j, \tilde{h}_{i_j}\right)$ be returned by oneshot Report Noisy Min applied to $(\mathcal{D}, h_1, \ldots, h_m)$.
3:    Set $h_{i_j} \equiv +\infty$.
4: **end for**

**Output:** indices $i_1, \ldots, i_k$ and approximations to $h_{i_1}(\mathcal{D}), \ldots, h_{i_k}(\mathcal{D})$

---

Therefore, for characterizing the degree of privacy preservation of Algorithm 2, the optimal DP composition theorem of EMs are required and stated in the following Lemma 3.

**Lemma 3.** *(Dong et al., 2020) If $\mathcal{M}$ is a $k$-fold non-adaptive composition of $\varepsilon$-BR mechanisms, then it is $\left(\varepsilon_g, \delta_k^{EM}(\varepsilon_g)\right)$-DP with*

$$\delta_k^{EM}(\varepsilon_g) = \max_{0 \le \ell \le k} \sum_{i=0}^{k} \binom{k}{i} p_{t_\ell^*}^{k-i} \left(1 - p_{t_\ell^*}\right)^i \left(e^{kt_\ell^* - i\varepsilon} - e^{\varepsilon_g}\right)_+,$$

*where $(a)_+$ is defined as $\max\{a, 0\}$, $p_t = \frac{e^{-t} - e^{-\varepsilon}}{1 - e^{-\varepsilon}}$ and $t_\ell^* = \frac{\varepsilon_g + (\ell+1)\varepsilon}{k+1}$ where if $t_\ell^* \notin [0, \varepsilon]$, then we round it to the closest point in $[0, \varepsilon]$.*

**Theorem 4.** *If $\{h_i\}_{i=1}^k$ are consistent, then Algorithm 2 ensures $(\varepsilon_1 + \varepsilon_2, \delta_k(\varepsilon_1) + \delta_k^{EM}(\varepsilon_2))$-DP for all $\varepsilon_1, \varepsilon_2 > 0$, where the expressions for $\delta_k(\varepsilon_1)$ and $\delta_k^{EM}(\varepsilon_2)$ are respectively given in Theorem 2 and Lemma 3 in which $\mu = \varepsilon = \frac{\Delta}{\gamma}$.*

### 3.2 PRIVACY-UTILITY PERFORMANCE COMPARISON

The most intuitive way to analyze the privacy-utility performance of the private top-$k$ algorithm is to compare the variance of the added noise. Under the same privacy guarantee, a smaller noise variance indicates that the output values are closer to the true values, and also signifies the higher privay-utility performance. Therefore, in this subsection, for the peeling algorithm under RNM with Laplace or Gaussian mechanim and the peeling algorithm under oneshot RNM with Gumbel mechanism, we compare the corresponding noise variances of Laplace, Gaussian and Gumbel mechanisms in these algorithms.

To ensure fairness in comparison, let the peeling algorithm under RNM with Laplace, Gaussian mechanism and the peeling algorithm under oneshot RNM satisfy $(\varepsilon, \delta)$-DP separately in private top-$k$ selection. By formulating the following optimization problems, we obtain the minimum noise variance corresponding to several algorithms. The peeling algorithm under RNM with $\text{Lap}\left(0, \frac{\Delta\sqrt{10k \ln(1/\delta)}}{\varepsilon}\right)$ is $(\varepsilon, \delta)$-DP (Dwork et al., 2021). The variance of Laplace distribution in the peeling algorithm under RNM is $\frac{\varepsilon^2}{5k\Delta^2 \ln(1/\delta)}$. Meanwhile, combined with the result in Cai et al. (2024), the peeling algorithm under RNM with $\mathcal{N}(0, \sigma^2)$ is $(\varepsilon, \delta)$-DP where the variance of Gaussian distribution $\sigma^2$ satisfies

$$\min_{0 < \varepsilon_0 < \varepsilon} \quad \sigma^2$$
$$\text{s.t.} \quad \delta_{\text{Gauss}}(\varepsilon_0) \le \delta,$$

where $\delta_{\text{Gauss}}(\varepsilon_0)$ is provided in Corollary 1 of Dong et al. (2022) with $\mu = \frac{\sqrt{8k}\Delta}{\sigma}$. Lastly, for Gumbel mechanism, Algorithm 2 is $(\varepsilon, \delta)$-DP utilizing Theorem 4 if the variance of Gumbel distribution

$\frac{\pi^2}{6}\gamma^2$ is satisfied that

$$\min_{\varepsilon_1, \varepsilon_2 > 0} \quad \frac{\pi^2}{6}\gamma^2$$
$$\text{s.t.} \quad \varepsilon_1 + \varepsilon_2 \leq \varepsilon,$$
$$\delta_k(\varepsilon_1) + \delta_k^{\text{EM}}(\varepsilon_2) \leq \delta,$$

where $\delta_k(\varepsilon_1)$ and $\delta_k^{\text{EM}}(\varepsilon_2)$ are respectively illustrated in Corollary 2 and Lemma 3 in which $\mu = \varepsilon = \frac{\Delta}{\gamma}$. Based on the above results, as shown in Fig. 3, by comparing at the same level of privacy protection, i.e., $(\varepsilon, \delta)$-DP, the noise variances of Gumbel mechanism are smaller than those of both Laplace and Gaussian mechanisms which also implies that the application of the Gumbel mechanism offers superior privacy-utility performance.

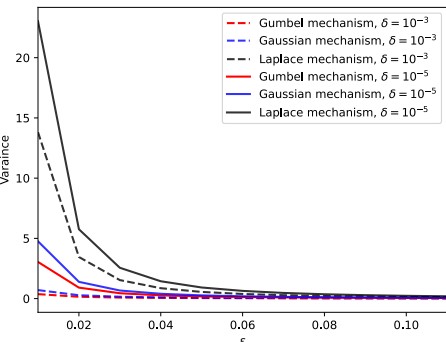

Figure 3: Noise variances comparison of Gaussian, Laplace mechanisms in the peeling algorithms under RNM and Gumbel mechanism in the peeling algorithm under oneshot RNM with $\varepsilon$ varying, $k = 10$ and $\delta = 10^{-3}, 10^{-5}$.

## 4 CONCLUSION

In this paper, we provide a different privacy-preserving top-$k$ selection algorithm with Gumbel mechanism, i.e., the peeling algorithm under oneshot RNM. Exploiting two special composition properties of the Gumbel mechanism, the oneshot RNM algorithm is designed, which is more efficient than the previous one as an algorithm that outputs the index and its query value without re-noising the query. To better characterize the privacy upper bound for the composition of $k$ Gumbel mechanisms hidden within the peeling algorithm, GumDP is presented in this work as a novel family of $f$-DPs. The $\mu$-GumDP analytically and tightly characterize the privacy of a single Gumbel mechanism, while the $(k, \mu)$-GumDP is presented as an extension to characterize the composition of $k$ Gumbel mechanisms under the assumption of consistency. To fairly compare different private top-$k$ selection algorithms, two equivalent transformation relationships between GumDP and DP are provided. Based on the above equivalence relations, the variance-based comparison shows that the new Gumbel-based algorithm outperforms the original Laplace- and Gaussian-based algorithms under the same privacy guarantees. It is evident that the Gumbel mechanism holds advantages as compared to the Gaussian and Laplace mechanisms in privacy-preserving selection algorithms.

Due to the wide range of practical applications involving top-$k$ selection algorithms, conducting in-depth and comprehensive research on private selection algorithms holds significant value. However, the topic of privacy selection still receives many challenges. In the process of extending the top-1 selection algorithm to the top-$k$ selection algorithm, only the peeling algorithm is studied in this work. The oneshot algorithm can be subsequently used to further improve the performance. Moreover, the composition of $k$ Gumbel mechanisms is taken under the strong assumption of consistency. Additionally, the practical application of this new private top-$k$ selection algorithm remains to be explored.

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

## A    PROOF OF EQUATION (4)

Let $\mathrm{Gum}(0,1)$ and $\mathrm{Gum}(\mu,1)$ be the distributions of $\mathcal{M}_{\mathrm{Gum}}(\mathcal{D})$ and $\mathcal{M}_{\mathrm{Gum}}(\mathcal{D}')$ in (2) respectively, and $p_0$ and $p_1$ be the PDFs of $\mathrm{Gum}(0,1)$ and $\mathrm{Gum}(\mu,1)$ respectively. For the hypothesis testing problem (2), the likelihood ratio is

$$\frac{p_1(x)}{p_0(x)} = \frac{e^{x-\mu-e^{x-\mu}}}{e^{x-e^x}} = e^{-\mu+e^x(1-e^{-\mu})},$$

which is a monotone increasing function in $x$. Thus, the rejection domain in (2) is $W = \{X > t\}$ where $X$ is random sample from Gumbel distribution and $t \in \mathcal{R}$. The corresponding type I and type II errors are

$$\alpha(t) = P(X > t | X \sim \mathrm{Gum}(0,1)) = e^{-e^t},$$

$$\beta(t) = P(X < t | X \sim \mathrm{Gum}(\mu,1)) = 1 - e^{-e^{t-\mu}}.$$

Solving $\alpha(t) = \alpha$ yields $t = \ln(-\ln\alpha)$. So

$$T(\mathrm{Gum}(0,1), \mathrm{Gum}(\mu,1))(\alpha) = 1 - e^{-e^{-\mu}(-\ln\alpha)} = 1 - (\alpha)^{e^{-\mu}}.$$

And

$$T(\mathrm{Gum}(\mu,1), \mathrm{Gum}(0,1))(\alpha) = T(\mathrm{Gum}(0,1), \mathrm{Gum}(\mu,1))^{-1}(\alpha) = (1-\alpha)^{e^{\mu}}.$$

Let $\alpha_1$ be unique solution of $T(\mathrm{Gum}(0,1), \mathrm{Gum}(\mu,1))'(\alpha) = -1$ and $\alpha_2 = T(\mathrm{Gum}(0,1), \mathrm{Gum}(\mu,1))(\alpha_1)$. Then, $\alpha_1 = e^{\frac{\mu}{e^{-\mu}-1}}, \alpha_2 = 1 - e^{\frac{\mu e^{-\mu}}{e^{-\mu}-1}}$. Similar to Eq.(13) in Dong et al. (2022),

$$B_\mu(\alpha) = \min\{T(\mathrm{Gum}(0,1), \mathrm{Gum}(\mu,1)), T(\mathrm{Gum}(\mu,1), \mathrm{Gum}(0,1))\}^{**}$$

$$= \begin{cases} T(\mathrm{Gum}(0,1), \mathrm{Gum}(\mu,1))(\alpha), & \alpha \in [0, \alpha_1), \\ \alpha_1 - \alpha + T(\mathrm{Gum}(0,1), \mathrm{Gum}(\mu,1))(\alpha_1), & \alpha \in [\alpha_1, \alpha_2), \\ T(\mathrm{Gum}(\mu,1), \mathrm{Gum}(0,1))(\alpha), & \alpha \in [\alpha_2, 1]. \end{cases}$$

## B    PROOF OF THEOREM 1

For any two neighboring databases $\mathcal{D}$ and $\mathcal{D}'$ and $\gamma \geq \Delta h/\mu$, we get

$$T(\mathcal{M}_{\mathrm{Gum}}(\mathcal{D}), \mathcal{M}_{\mathrm{Gum}}(\mathcal{D}')) = T(\mathrm{Gum}(h(\mathcal{D}), \gamma), \mathrm{Gum}(h(\mathcal{D}'), \gamma))$$

$$\geq \min\{T(\mathrm{Gum}(0,1), \mathrm{Gum}(|h(\mathcal{D}) - h(\mathcal{D}')|/\gamma, 1)),$$

$$T(\mathrm{Gum}(|h(\mathcal{D}) - h(\mathcal{D}')|/\gamma, 1), \mathrm{Gum}(0,1))\}$$

$$\geq B_{\frac{|h(\mathcal{D})-h(\mathcal{D}')|}{\gamma}}.$$

By the definition of sensitivity, $|h(\mathcal{D}) - h(\mathcal{D}')| \leq \Delta h \leq \gamma\mu$. Therefore, we get

$$T(\mathcal{M}_{\mathrm{Gum}}(\mathcal{D}), \mathcal{M}_{\mathrm{Gum}}(\mathcal{D}')) \geq B_{\frac{|h(\mathcal{D})-h(\mathcal{D}')|}{\gamma}} \geq B_\mu.$$

## C    PROOF OF COROLLARY 1

Based on the equivalent conversion of $f$-DP and DP and the symmetry of the function $B_\mu$, $\mu$-GumDP is equal to $(\varepsilon, 1 + B_\mu^*(-e^\varepsilon))$-DP. Therefore, we only need to compute the $B_\mu^*(-e^\varepsilon)$. From the definition of convex conjugate function, $B_\mu^*(y) = \sup_{x \in [0,1]}(yx - B_\mu(x))$. And, from the shape of $B_\mu$, the supremum is obtained only at the unique critical point when $y \in (-\infty, -1)$. From

$$0 = \frac{\mathrm{d}}{\mathrm{d}x}(yx - B_\mu(x)) = \frac{\mathrm{d}}{\mathrm{d}x}(yx - 1 + x^{e^{-\mu}})$$

$$= y + e^{-\mu}x^{e^{-\mu}-1},$$

we have $x = (-e^\mu y)^{\frac{1}{e^{-\mu}-1}}$. Then,

$$B_\mu^*(y) = y(-e^\mu y)^{\frac{1}{e^{-\mu}-1}} + (-e^\mu y)^{\frac{e^{-\mu}}{e^{-\mu}-1}} - 1, \qquad y \in (-\infty, -1).$$

Setting $y = -e^\varepsilon$ implies $B_\mu^*(-e^\varepsilon) = (e^{\mu+\varepsilon} - e^\varepsilon)e^{\frac{\mu+\varepsilon}{e^{-\mu}-1}} - 1$. Thus, this corollary holds.

# D  PROOF OF EQUATION (7)

Let $\text{Gum}(0,1)$ and $\text{Gum}(\mu,1)$ be the distributions of $\mathcal{M}_{\text{Gum}}(\mathcal{D})$ and $\mathcal{M}_{\text{Gum}}(\mathcal{D}')$ in (6) respectively, and $p_0$ and $p_1$ be the PDFs of $(y_1, y_2, \ldots, y_k)$ under $H_0$ and $H_1$ respectively. For the hypothesis testing problem (6), the likelihood ratio is

$$\frac{p_1(x_1, x_2, \ldots, x_k)}{p_0(x_1, x_2, \ldots, x_k)} = \prod_{i=1}^{k} \frac{e^{(x_i-\mu)-e^{(x_i-\mu)}}}{e^{x_i-e^{x_i}}} = e^{-k\mu} e^{(1-e^{-\mu})\sum_{i=1}^{k} e^{x_i}}.$$

It is a monotonically increasing function in $\sum_{i=1}^{k} e^{\frac{x_i}{\beta}}$. Thus, the rejection domain in (6) is $W = \{\sum_{i=1}^{k} e^{\frac{X_i}{\beta}} > t\}$ where $X_i$ is a random sample and $t > 0$. The corresponding type I and type II errors respectively are

$$\alpha(t) = P\left(\sum_{i=1}^{k} e^{X_i} > t \,\Big|\, \{X_i\}_{i=1}^{k} \overset{\text{i.i.d.}}{\sim} \text{Gum}(0,1)\right), \tag{8}$$

$$\beta(t) = P\left(\sum_{i=1}^{k} e^{X_i} < t \,\Big|\, \{X_i\}_{i=1}^{k} \overset{\text{i.i.d.}}{\sim} \text{Gum}(\mu,1)\right). \tag{9}$$

To facilitate the analysis, let $y_i = e^{x_i}, i = 1, 2, \ldots, k$. When $X_i \sim \text{Gum}(0,1)$, $P_{Y_i}(y_i \leq t) = P_{X_i}(e^{x_i} \leq t) = P_{X_i}(x_i \leq \ln t) = 1 - e^{-t}$, the distribution of $Y_i$ is the exponential distribution with parameter 1, denoted as $\text{Exp}(1)$. Since $\{X_i\}_{i=1}^{k}$ are distributed independently and identically, so are $\{Y_i\}_{i=1}^{k}$. Based on the nature of the exponential distribution, $\sum_{i=1}^{k} Y_i \sim \Gamma(k,1)$ where $\Gamma(k,1)$ denotes the Gamma distribution with shape parameter $k$ and inverse scale parameter 1. Similarly, when $X_i \sim \text{Gum}(\mu,1)$, $P_{Y_i}(y_i \leq t) = 1 - e^{-e^{\ln t - \mu}} = 1 - e^{-e^{-\mu}t}$, so $Y_i \sim \text{Exp}(e^{-\mu})$ and $\sum_{i=1}^{k} Y_i \sim \Gamma(k, e^{-\mu})$. Then, (8) and (9) respectively become

$$\alpha(t) = P\left(\sum_{i=1}^{k} Y_i > t \,\Big|\, \{Y_i\}_{i=1}^{k} \overset{\text{i.i.d.}}{\sim} \text{Exp}(1)\right)$$

$$= P(\xi > t | \xi \sim \Gamma(k,1))$$

$$= e^{-t}\left(1 + \sum_{i=1}^{k-1} \frac{t^i}{i!}\right),$$

$$\beta(t) = P\left(\sum_{i=1}^{k} Y_i < t | \{Y_i\}_{i=1}^{k} \overset{\text{i.i.d.}}{\sim} \text{Exp}(e^{-\mu})\right)$$

$$= P(\xi < t | \xi \sim \Gamma(k, e^{-\mu}))$$

$$= 1 - e^{-te^{-\mu}}\left(1 + \sum_{i=1}^{k-1} \frac{(te^{-\mu})^i}{i!}\right).$$

Due to $Y \sim \chi^2(2k)$, $F_Y(x) = 1 - e^{-\frac{x}{2}}\left(1 + \sum_{i=1}^{k-1} \frac{(\frac{x}{2})^i}{i!}\right)$. So, $\alpha(t) = 1 - F_Y(2t)$ and $\beta(t) = F_Y\left(\frac{1}{2}F_Y^{-1}(1-\alpha(t))2e^{-\mu}\right) = F_Y\left(F_Y^{-1}(1-\alpha(t))e^{-\mu}\right)$, which yields

$$T(\text{Gum}(0,1)^k, \text{Gum}(\mu,1)^k)(\alpha) = F_Y\left(F_Y^{-1}(1-\alpha)e^{-\mu}\right).$$

Analogously, under the original hypothesis obeying $\text{Gum}(\mu,1)^k$ and the alternative hypothesis obeying $\text{Gum}(0,1)^k$, the type I and type II errors are respectively $\alpha(t) = 1 - e^{-te^{-\mu}}\left(1 + \sum_{i=1}^{k-1} \frac{(te^{-\mu})^i}{i!}\right)$ and $\beta(t) = e^{-t}\left(1 + \sum_{i=1}^{k-1} \frac{t^i}{i!}\right)$. Easily obtained, $\alpha(t) = F_Y(2e^{-\mu}t)$ and $\beta(t) = 1 - F_Y\left(2F_Y^{-1}(\alpha(t))\frac{1}{2}e^{\mu}\right) = 1 - F_Y\left(F_Y^{-1}(\alpha(t))e^{\mu}\right)$, which yields

$$T(\text{Gum}(\mu,1)^k, \text{Gum}(0,1)^k)(\alpha) = 1 - F_Y\left(F_Y^{-1}(\alpha(t))e^{\mu}\right).$$

Being identical to the proof of Equation (4), let $\alpha_1$ be unique solution of $T(\mathrm{Gum}(0,1)^k, \mathrm{Gum}(\mu,1)^k)'(\alpha) = -1$ and $\alpha_2 = T(\mathrm{Gum}(0,1)^k, \mathrm{Gum}(\mu,1)^k)(\alpha_1)$. Taking the derivative of $T(\mathrm{Gum}(0,1)^k, \mathrm{Gum}(\mu,1)^k)(\alpha)$ and setting it to $-1$ yields

$$
-1 = \frac{\mathrm{d}}{\mathrm{d}\alpha} T(\mathrm{Gum}(0,1)^k, \mathrm{Gum}(\mu,1)^k)(\alpha) = \frac{\mathrm{d}}{\mathrm{d}\alpha} F_Y\left(F_Y^{-1}(1-\alpha)e^{-\mu}\right)
$$

$$
= \frac{-p_Y\left(F_Y^{-1}(1-\alpha)e^{-\mu}\right)e^{-\mu}}{p_Y\left(F_Y^{-1}(1-\alpha)\right)}
$$

$$
= -e^{\frac{1}{2}F_Y^{-1}(1-\alpha)\left(1-e^{-\mu}\right)-k\mu},
$$

where $p_Y$ is the PDF of $\chi^2(2k)$. Then, $\alpha_1 = 1 - F_Y\left(\frac{2k\mu}{1-e^{-\mu}}\right)$ and $\alpha_2 = F_Y(F_Y^{-1}(1-\alpha_1)e^{-\mu}) = F_Y\left(\frac{2k\mu e^{-\mu}}{1-e^{-\mu}}\right)$. This proof is finished.

## E  PROOF OF THEOREM 2

Because of the consistence of $\{h_i\}_{i=1}^k$,

$T(\mathcal{M}(\mathcal{D}), \mathcal{M}(\mathcal{D}'))$

$= T(\mathrm{Gum}(h_1(\mathcal{D}),\gamma) \times \cdots \times \mathrm{Gum}(h_k(\mathcal{D}),\gamma), \mathrm{Gum}(h_1(\mathcal{D}'),\gamma) \times \cdots \times \mathrm{Gum}(h_k(\mathcal{D}'),\gamma))$

$= T(\mathrm{Gum}(0,1)^k, \mathrm{Gum}((h_1(\mathcal{D}')-h_1(\mathcal{D}))/\gamma,1) \times \cdots \times \mathrm{Gum}((h_k(\mathcal{D}')-h_k(\mathcal{D}))/\gamma,1))$

$\geq T(\mathrm{Gum}(0,1)^k, \mathrm{Gum}(\mathrm{sign}(h_1(\mathcal{D}')-h_1(\mathcal{D}))\mu,1) \times \cdots \times \mathrm{Gum}(\mathrm{sign}(h_k(\mathcal{D}')-h_k(\mathcal{D}))\mu,1))$

$\geq \min\{T(\mathrm{Gum}(0,1)^k, \mathrm{Gum}(\mu,1)^k), T(\mathrm{Gum}(\mu,1)^k, \mathrm{Gum}(0,1)^k)\}$

$\geq B_\mu^k.$

The proof is completed.

## F  PROOF OF COROLLARY 2

Similarly to the proof of Theorem 1, by the symmetry of the function $B_\mu^k$, $(k,\mu)$-GumDP is equal to $(\varepsilon, 1 + B_\mu^{k^*}(-e^\varepsilon))$-DP. Therefore, we only need to compute the $B_\mu^{k^*}(-e^\varepsilon)$. Before that we need to know that the PDF of $Z$ is

$$
p_Z(x) = \begin{cases} \frac{1}{2^k\Gamma(k)}x^{k-1}e^{-\frac{x}{2}}, & x > 0, \\ 0, & \text{otherwise.} \end{cases}
$$

It is easy to get $B_\mu^{k^*}(y) = \sup_{x\in[0,1]}(yx - B_\mu^k(x))$. And, from the shape of $B_\mu^k$, the supremum is obtained only at the unique critical point when $y \in (-\infty, -1)$. Taking the derivative of the objective function and setting it to zero yields, when $y \in (-\infty, -1)$,

$$
0 = \frac{\mathrm{d}}{\mathrm{d}x}(yx - B_\mu^k(x)) = \frac{\mathrm{d}}{\mathrm{d}x}(yx - F_Z(F_Z^{-1}(1-x)e^{-\mu})
$$

$$
= y + \frac{e^{-\mu}p_Z(F_Z^{-1}(1-x)e^{-\mu})}{p_Z(F_Z^{-1}(1-x))}
$$

$$
= y + e^{\frac{F_Z^{-1}(1-x)}{2}(1-e^{-\mu})-k\mu}.
$$

Getting $x = 1 - F_Z\left(\frac{2(k\mu+\ln(-y))}{1-e^{-\mu}}\right)$ and taking it into $B_\mu^{k^*}$ leads to

$$
B_\mu^{k^*}(x) = y\left(1 - F_Z\left(\frac{2(k\mu+\ln(-y))}{1-e^{-\mu}}\right)\right) - F_Z\left(F_Z^{-1}\left(F_Z\left(\frac{2(k\mu+\ln(-y))}{1-e^{-\mu}}\right)\right)e^{-\mu}\right)
$$

$$
= y - yF_Z\left(\frac{2(k\mu+\ln(-y))}{1-e^{-\mu}}\right) - F_Z\left(\frac{2(k\mu+\ln(-y))e^{-\mu}}{1-e^{-\mu}}\right), \qquad y \in (-\infty, -1).
$$

When $y = -e^\varepsilon$, $B_\mu^{k^*}(-e^\varepsilon) = -e^\varepsilon + e^\varepsilon F_Z\left(\frac{2(k\mu+\varepsilon)}{1-e^{-\mu}}\right) - F_Z\left(\frac{2(k\mu+\varepsilon)e^{-\mu}}{1-e^{-\mu}}\right).$

## G PROOF OF LEMMA 1

Start by analyzing the distribution of $\mathcal{M}^m_{\text{minGum}}(\mathcal{D})$, if $\{\eta_i\}^m_{i=1} \stackrel{\text{i.i.d.}}{\sim} \text{Gum}(0, \frac{\Delta}{\varepsilon})$,

$$P\left(\mathcal{M}^m_{\text{minGum}}(\mathcal{D}) > t\right) = P\left(\min_{i\in[m]}\left\{\mathcal{M}^{(i)}_{\text{Gum}}(\mathcal{D})\right\}\right)$$

$$= P\left(\min_{i\in[m]}\{h_i(\mathcal{D}) + \eta_i\} > t\right)$$

$$= \prod_{i=1}^m P(h_i(\mathcal{D}) + \eta_i > t)$$

$$= \prod_{i=1}^m e^{-e^{\frac{t-h_i(\mathcal{D})}{\frac{\Delta}{\varepsilon}}}}$$

$$= e^{-e^{\frac{t+\frac{\Delta}{\varepsilon}\ln\left(\sum_{i=1}^m e^{-\frac{h_i(\mathcal{D})}{\frac{\Delta}{\varepsilon}}}\right)}{\frac{\Delta}{\varepsilon}}}}.$$

It is easy to see that $\mathcal{M}^m_{\text{minGum}}$ as a Gumbel mechanism and $\mathcal{M}^m_{\text{minGum}}(\mathcal{D})$ distributed from

$$\text{Gum}\left(-\frac{\Delta}{\varepsilon}\ln\left(\sum_{i=1}^m e^{-\frac{h_i(\mathcal{D})}{\frac{\Delta}{\varepsilon}}}\right), \frac{\Delta}{\varepsilon}\right).$$

Let $g(\mathcal{D}) = -\frac{\Delta}{\varepsilon}\ln\left(\sum_{i=1}^m e^{-\frac{h_i(\mathcal{D})}{\frac{\Delta}{\varepsilon}}}\right)$. Then, $\mathcal{M}^m_{\text{minGum}}(\mathcal{D}) = g(\mathcal{D}) + \eta$ where $\eta \sim \text{Gum}(0, \frac{\Delta}{\varepsilon})$. Because of $\Delta = \max_{i\in[k]}\max_{\mathcal{D},\mathcal{D}'}|h_i(\mathcal{D}) - h_i(\mathcal{D}')|$, $|h_i(\mathcal{D}) - h_i(\mathcal{D}')| \leq \Delta$. So,

$$e^{-\varepsilon}\sum_{i=1}^m e^{-\frac{h_i(\mathcal{D})}{\frac{\Delta}{\varepsilon}}} \leq \sum_{i=1}^m e^{-\frac{h_i(\mathcal{D}')}{\frac{\Delta}{\varepsilon}}} \leq e^{\varepsilon}\sum_{i=1}^m e^{-\frac{h_i(\mathcal{D})}{\frac{\Delta}{\varepsilon}}}.$$

Then, for any $i \in [k]$,

$$|g(\mathcal{D}) - g(\mathcal{D}')| = \left|-\frac{\Delta}{\varepsilon}\ln\left(\sum_{i=1}^m e^{-\frac{h_i(\mathcal{D})}{\frac{\Delta}{\varepsilon}}}\right) + \frac{\Delta}{\varepsilon}\ln\left(\sum_{i=1}^m e^{-\frac{h_i(\mathcal{D}')}{\frac{\Delta}{\varepsilon}}}\right)\right|$$

$$= \left|\frac{\Delta}{\varepsilon}\ln\left(\frac{\sum_{i=1}^m e^{-\frac{h_i(\mathcal{D}')}{\frac{\Delta}{\varepsilon}}}}{\sum_{i=1}^m e^{-\frac{h_i(\mathcal{D})}{\frac{\Delta}{\varepsilon}}}}\right)\right| \leq \left|\frac{\Delta}{\varepsilon}\ln e^{\varepsilon}\right| = \Delta.$$

So $|g(\mathcal{D}) - g(\mathcal{D}')| \leq \Delta$ holds in the general case. Because of $\max_{\mathcal{D},\mathcal{D}'}|g(\mathcal{D}) - g(\mathcal{D}')| \leq \Delta$, $\mathcal{M}^m_{\text{minGum}}$ satisfies $\mu$-GumDP using Theorem 2.

## H PROOF OF LEMMA 2

Let $\{\eta_j\}^m_{j=1}$ be i.i.d.copied from $\text{Gum}(0, \frac{\Delta}{\varepsilon})$ and $\mathcal{M}^{(j)}_{\text{Gum}}(\mathcal{D}) = h_j(\mathcal{D}) + \eta_j$, $j \in [m]$. Then, for any $i \in [m]$ and $t \in \mathcal{R}$,

$$P\left(M^*_{\text{Gum}}(D) = (i, h_i(D) + \eta_i) \in [m] \times (t, +\infty)\right)$$

$$= \int_t^\infty p\left(u_i - h_i(D), 0, \frac{\Delta}{\varepsilon}\right)\prod_{j\in[m]\setminus\{i\}}\left(1 - F\left(u_i - h_j(D), 0, \frac{\Delta}{\varepsilon}\right)\right)du_i$$

$$= e^{-\sum_{j=1}^m e^{\frac{\varepsilon(t-h_j(D))}{\Delta}}} \cdot \frac{e^{\frac{-\varepsilon h_i(D)}{\Delta}}}{\sum_{j=1}^m e^{\frac{-\varepsilon h_j(D)}{\Delta}}}$$

$$= P\left(\min_{j\in[m]}\left\{\mathcal{M}^{(j)}_{\text{Gum}}(\mathcal{D})\right\} > t\right) \cdot P\left(\mathcal{M}_{\text{E}}(\mathcal{D}, \{h_j\}_{j\in[m]}, \varepsilon) = i\right)$$

$$= P\left(\mathcal{M}^m_{\text{minGum}}(\mathcal{D}) > t\right) \cdot P\left(\mathcal{M}_{\text{E}}(\mathcal{D}, \{h_j\}_{j\in[m]}, \varepsilon) = i\right),$$

where $p$ and $F$ are the PDF and CDF of $\mathrm{Gum}(0, \frac{\Delta}{\varepsilon})$ respectively. With Lemma 1, the proof is complete.

## I  PROOF OF THEOREM 3

By Lemma 2, Algorithm 1 is equivalent to the independent composition of a mechanism satisfied $\frac{\Delta}{\gamma}$-GumDP and the EM $\mathcal{M}_{\mathrm{E}} \left( \mathcal{D}, \{h_j\}_{j \in [m]}, \frac{\Delta}{\gamma} \right)$. Because of the monotone, $\mathcal{M}_{\mathrm{E}}$ is $\left( \frac{\Delta}{\gamma}, 0 \right)$-DP. And, due to Theorem 1 and the basic composition theorem in Dwork & Roth (2013), the algorithm is $\left( \frac{\Delta}{\gamma} + \varepsilon, \delta(\varepsilon) \right)$-DP where $\delta(\varepsilon) = \left( e^{\frac{\Delta}{\gamma} + \varepsilon} - e^{\varepsilon} \right) e^{e^{\frac{\frac{\Delta}{\gamma} + \varepsilon}{-\frac{\Delta}{\gamma} - 1}}}$ for any $\varepsilon > 0$.

## J  PROOF OF THEOREM 4

Based on the result in Dong et al. (2020), the $\varepsilon$-BR is equal to $\varepsilon$-DP when the functions $\{h_i\}_{i \in [m]}$ are monotone. Similarly to Theorem 3, it can be proved by Theorem 2, Lemma 3 and the basic composition theorem.

