# OpenReview forum: "Private Top-$k$ Selection under Gumbel Differential Privacy Guarantees"
_ICLR.cc/2026/Conference — ICLR 2026 Conference Withdrawn Submission_

### Official Review · Reviewer_3Fit · 2025-10-24

**Soundness:** 2
**Presentation:** 1
**Contribution:** 2
**Rating:** 2
**Confidence:** 4

**Summary:**

This work considers joint release of max and argmax values of a set of functions evaluations on some dataset in a differentially private manner, that is, releasing both the index and the value of the function achieving the minimal (maximal) evaluation, and its extension to top-k functions. For a single index, this is typically achieved by first identifying the maximal index in a differentially private manner using the report-noisy-max algorithm, followed by a noisy evaluation of the query in that index. Extension to top-k is achieved using the peeling algorithm which sequentially removes the maximal index.

One possible implementation of the report noisy max algorithm is by adding Gumbel noise to each functions evaluation, and reporting the index of the noisy max. This work considers an alternative algorithm which rather than releasing a separate evaluation of the function in the reported index, simply uses the original noisy evaluation. To analyze this option the authors utilize the f-DP framework by computing the explicit tradeoff function of the Gumbel mechanism.

**Strengths:**

The work is well motivated, clearly structured, and its results seem plausible.

**Weaknesses:**

Unfortunately, it seems like there are many incorrect / inaccurate details in the write-up, which - while all might be fixable, make it nearly impossible to evaluate in its current form. The fact this work does not even contain proof outlines for most claims, makes it even harder to evaluate.

1. The CDF of the Gumbel distribution is $e^{-e^{(x-\mu)/\gamma}}$, not $1-e^{-e^{(x-\mu)/\gamma}}$ as stated in the opening of Section 2.1. This is not a simple typo, but seems to affect the analysis of some of the proofs (e.g., Lemmas 1 and 2). It might very well be the case that switching CDF and CCDF in the appropriate places will solve the issue, but it its current form it makes it hard to follow some of the proofs, e.g., Lemma 2 discussed below.
2. If I understand correctly, Lemma 1 is incorrect in its current for, and requires an additional assumption to hold. It's proof relies of the claim (last line of the proof) that $\vert g(\mathcal{D}) - g(\mathcal{D}') \vert \le \Delta$ but in the general case it is in fact $\Delta + \frac{\Delta}{\epsilon}\ln(m)$. The $\ln(m)$ term can be ignored only under the additional assumption that changing a single element can change the evaluation of only a single function.While this assumption indeed holds for some functions such as histograms, it does not hold in general.
3. Lemma 2 is started in a way that is very hard to follow. Assuming I understood correctly, and the claim is that releasing jointly the maximal index and value of the Gumbel mechanism is distributed identically like independent evaluation of the maximal index and value, I fail to understand the proof, specifically the second equality. This might also have to do with the CDF/CCDF issue mentioned before, but regardless at least one of the two terms must take the form of $p(t)$ and the other $1-p(t)$, which I can't find.

Besides these clarity issues, I further fail to understand how the two parts of this work are related to each other. Even if Lemma 2 is correct, it essentially allows reducing the analysis of $M^{*}_{Gum}$ to the composition of the Gumbel noise addition mechanism and the exponential mechanism, which can be done using basic composition. the privacy guarantees of the Gumbel noise addition mechanism can be analyze directly via DP definition.

Furthermore, the It is not clear if the Gumbel mechanism is even a useful tool in most cases. Using corollary 1 it is clear that for somewhat large $\mu$ (say $\approx5$), $\delta(\varepsilon)$ approaches $1$.

Indeed, Figure 3 shows that the advantage of the Gumbel mechanism relative to the Gaussian mechanism is negligible except for extremely small values of $\varepsilon$.

**Questions:**

Please address the questions above.

---

> ### Author Response · Authors · 2025-11-17
>
> We thank the reviewer for insightful comments. We have carefully considered your comments and responded to the individual concerns. To help you identify the changes in the revised version, we have hightlight all changes in blue.
>
> Q1: The CDF of the Gumbel distribution is \\(e^{-e^{(x-\\mu) / \\gamma}}\\), not \\(1-e^{-e^{(x-\\mu) / \\gamma}}\\) as stated in the opening of Section 2.1. This is not a simple typo, but seems to affect the analysis of some of the proofs (e.g., Lemmas 1 and 2). It might very well be the case that switching CDF and CCDF in the appropriate places will solve the issue, but it its current form it makes it hard to follow some of the proofs, e.g., Lemma 2 discussed below.
>
> A1:  Thank you for pointing this out. The distribution used in our analysis is the minimum Gumbel distribution. For  \\(X \\sim \\operatorname{Gumbel}\_{\\min}(\\mu,\\gamma)\\) with \\(\\gamma>0\\), the density is \\(p(x;\\mu,\\gamma)=\\frac{1}{\\gamma}\\exp\\left(\\frac{x-\\mu}{\\gamma}-e^{\\frac{x-\\mu}{\\gamma}}\\right),\\) and hence the CDF is \\(F(x)=\int_{-\infty}^{x} p(y;\mu,\gamma)dy =1-\exp\left(-e^{\frac{x-\mu}{\gamma}}\right).\\) Thus, the CDF stated in the paper is correct for the minimum Gumbel distribution. We appreciate that this distinction was not sufficiently emphasized, and we have revised the exposition for clarity on Page 4 of the revision.
>
> Q2： If I understand correctly, Lemma 1 is incorrect in its current for, and requires an additional assumption to hold. It's proof relies of the claim (last line of the proof) that \\(\left|g(\mathcal{D})-g\left(\mathcal{D}^{\prime}\right)\right| \leq \Delta\\) but in the general case it is in fact \\(\Delta+\frac{\Delta}{\epsilon} \ln (m)\\). The \\(\ln (m)\\) term can be ignored only under the additional assumption that changing a single element can change the evaluation of only a single function. While this assumption indeed holds for some functions such as histograms, it does not hold in general.
>
> A2： We thank the reviewer for raising this concern. Given \\(\Delta=\max_{i\in[k]}\max_{D,D'}|h_i(D)-h_i(D')|,\\) for all \\(i\\), \\(|h_i(D)-h_i(D^{\prime})|\le \Delta\\). Then, \\(e^{-\varepsilon} \sum_{i=1}^m \exp\left(-\frac{h_i(D)}{\Delta/\varepsilon}\right) \le \sum_{i=1}^m \exp\left(-\frac{h_i(D^{\prime})}{\Delta/\varepsilon}\right) \le e^{\varepsilon} \sum_{i=1}^m \exp\left(-\frac{h_i(D)}{\Delta/\varepsilon}\right).\\) For \\(g(D)=-\frac{\Delta}{\varepsilon} \ln\left(\sum_{i=1}^m \exp\left(-\frac{h_i(D)}{\Delta/\varepsilon}\right)\right),\\) we obtain \\(|g(D)-g(D^{\prime})|=\frac{\Delta}{\varepsilon}\Bigl|\ln\left(\frac{\sum_i e^{-h_i(D')/(\Delta/\varepsilon)}}{\sum_i e^{-h_i(D)/(\Delta/\varepsilon)}}\right)\Bigr|\le \frac{\Delta}{\varepsilon}\ln(e^{\varepsilon})=\Delta.\\) Thus, the claimed Lipschitz bound holds without additional assumptions. We have expanded the proof on Page 16 of the revised version for completeness.
>
> Q3：  Lemma 2 is started in a way that is very hard to follow. Assuming I understood correctly, and the claim is that releasing jointly the maximal index and value of the Gumbel mechanism is distributed identically like independent evaluation of the maximal index and value, I fail to understand the proof, specifically the second equality. This might also have to do with the CDF/CCDF issue mentioned before, but regardless at least one of the two terms must take the form of \\(p(t)\\) and the other \\(1-p(t)\\), which I can't find.
>
> A3： We appreciate the reviewer’s feedback on Lemma 2. The equality in question is correct under the CDF of the minimum Gumbel distribution used above. The lemma exploits the fact that adding independent minimum-Gumbel noise preserves the same ordering structure as independently selecting the minimum index and corresponding noised value. The discrepancy noted by the reviewer appears to stem from interpreting the distribution as a maximum Gumbel; we have clarified the setup and exposition to avoid confusion on Page 4.

---

> > ### Author Response · Authors · 2025-11-17
> >
> > Q4： Besides these clarity issues, I further fail to understand how the two parts of this work are related to each other. Even if Lemma 2 is correct, it essentially allows reducing the analysis of $M_{\text {Gum }}^*$ to the composition of the Gumbel noise addition mechanism and the exponential mechanism, which can be done using basic composition. the privacy guarantees of the Gumbel noise addition mechanism can be analyze directly via DP definition.
> >
> > A4： Regarding the connection between the two parts of the paper: although Lemma 2 ultimately enables representing the extended mechanism as a composition of (1) Gumbel noise addition on values and (2) an exponential mechanism on indices, the key point is that the extended procedure involves a non-independent composition. Standard \\(f\\)-DP or DP composition does not apply. Lemma 2 shows that, due to structural properties of the minimum Gumbel distribution, this non-independent mechanism is nevertheless equivalent to an independent composition—this is not a generic property of noise-additive mechanisms, and thus constitutes a main technical contribution.
> >
> > Q5： Furthermore, the It is not clear if the Gumbel mechanism is even a useful tool in most cases. Using corollary 1 it is clear that for somewhat large $\mu($ say $\approx 5), \delta(\varepsilon)$ approaches 1.
> >
> > A5：We agree that for large \\(\mu\\) the trade-off function degenerates. However, comparing with \\(\mu\\)-GDP illustrates that Gumbel noise is not inherently inferior. For example, at \\(\mu=5\\) and \\(\varepsilon=1\\), \\(\delta_{\mathrm{Gumbel}}(\varepsilon)\approx 0.9536, \delta_{\mathrm{Gauss}}(\varepsilon)\approx 0.9796,\\). Therefore, in practice we keep \\(\mu\\) small, consistent with standard practice for GDP as well. Thus, the observation that \\(\delta(\varepsilon)\\) approaches \\(1\\) for large \\(\mu\\) does not imply that the Gumbel mechanism is impractical.
> >
> > Q6： Indeed, Figure 3 shows that the advantage of the Gumbel mechanism relative to the Gaussian mechanism is negligible except for extremely small values of $\varepsilon$.
> >
> > A6： Thank you for raising this point.  You are correct. Regarding Figure3: indeed, the improvement over the Gaussian mechanism becomes pronounced primarily for very small $\varepsilon$. This corresponds to high-privacy regimes where tighter noise addition is crucial. In such settings, the Gumbel mechanism provides a nontrivial advantage, and our results identify this regime precisely.

---

> > > ### Comment · Reviewer_3Fit · 2025-11-19
> > >
> > > I thank the authors for their detailed response.
> > >
> > > * Q1-3 were sufficiently addressed
> > > * Q4 referred to the privacy derived by composing two independent mechanisms which seemingly should lead to similar bounds (and if not, it should be explains)
> > > * Q5 pointed out that for moderately large values of $\mu$ we have $\delta \approx 1$ *for any $\epsilon$* while other mechanisms have $\delta \ll 1$ for sufficiently large $\epsilon$.
> > >
> > > I think this work contains a meaningful contribution which is not approachable at this point, and hope the authors submit a revised version in the close future.

---

> > > > ### Author Response · Authors · 2025-12-02
> > > >
> > > > We sincerely thank the reviewer for the helpful feedback.  We have attempted to address each of your comments individually.
> > > >
> > > > Q4 referred to the privacy derived by composing two independent mechanisms which seemingly should lead to similar bounds (and if not, it should be explains)
> > > >
> > > > A4: It is equivalent to the combination of two independent mechanisms. However, our algorithm considers the joint output of the minimum value of several function evaluations on a given dataset and the argmin value, thereby avoiding quadratic noise addition to the minimum value. Here, we have uncovered that when the two mechanisms are combined in a correlated manner, it is equivalent to an independent combination. This is highly significant.
> > > >
> > > > Q5 pointed out that for moderately large values \\(\mu\\) of we have  \\(\delta\approx1\\) for any while other mechanisms have \\(\delta\ll1\\) for sufficiently large \\(\varepsilon\\).
> > > >
> > > > Q5: According Corollary 1, \\(\delta(\varepsilon)=\left(e^{\mu+\varepsilon}-e^{\varepsilon}\right) e^{\frac{\mu+\varepsilon}{e^{-\mu}-1}}= e^{-\frac{\varepsilon}{e^{\mu}-1}}\left(e^{\mu}-1\right) e^{\frac{\mu}{e^{-\mu}-1}}\\). So the propsed mechanism also has \\(\delta\ll1\\) for sufficiently large \\(\varepsilon\\).

---

### Official Review · Reviewer_8q7j · 2025-10-25

**Soundness:** 2
**Presentation:** 3
**Contribution:** 2
**Rating:** 2
**Confidence:** 4

**Summary:**

This paper introduces a new top-k selection algorithm based on the Gumbel mechanism under $f$-differential privacy ($f$-DP). The authors proposes a novel notion of privacy termed  $\mu$-Gumbel DP, which is a instantiation of $f$-DP obtained by specifying an appropriate trade-off function. Properties of $\mu$-Gumbel DP in the context of private selection problem are also presented. The authors also demonstrate better privacy-utility trade-off compared with existing methods via simulation experiments.

**Strengths:**

The paper offers a new perfective on privately solving the top-k selection problem, which is an important and  classic problem in differential privacy. I also find the composition properties presented in the paper interesting.

**Weaknesses:**

1. The applicability of the composition result shown in Theorem 2 is limited. Theorem 2 holds when all statistics $h_i$ are consistent, which is a rather strong assumption, especially for large $k$. Moreover, does the definition of consistency in Definition 9 need to hold for any neighboring dataset $D$ and $D'$? If that is the case, the practical use of Theorem 2 will be quite limited from my perspective. In addition, it would be helpful to clarify whether the consistency condition must be verified prior to applying the Gumbel mechanism within an algorithm.
2. Lemma 1 looks like a standard post-processing result to me. I am curious if Gumbel DP also satisfies post-processing property. If yes, is it a new property specific to Gumbel DP or property inherited from the general f-DP?
3. The utility metric shown in 3.2 is not entirely convincing. The authors use noise variance to analyze the privacy-utility performance. However, the private-k selection problem has a more common and meaningful metric that is the difference between the output statistics and the true minimum value. The absence of analytical results and comparison suing this metric significant weaken the contribution of this work.

**Questions:**

All questions are included in "Weaknesses" section.

---

> ### Author Response · Authors · 2025-11-17
>
> We thank the reviewer for insightful comments. We have carefully considered your comments and responded to the individual concerns.
>
> Q1：The applicability of the composition result shown in Theorem 2 is limited. Theorem 2 holds when all statistics \\(h_i\\) are consistent, which is a rather strong assumption, especially for large \\(k\\) . Moreover, does the definition of consistency in Definition 9 need to hold for any neighboring dataset \\(D\\) and \\(D^{\prime}\\)? If that is the case, the practical use of Theorem 2 will be quite limited from my perspective. In addition, it would be helpful to clarify whether the consistency condition must be verified prior to applying the Gumbel mechanism within an algorithm.
>
> A1：We appreciate the reviewer’s concerns. The original consistency assumption was indeed strong, and we agree that requiring it to hold for all neighbouring datasets limits applicability. In response, we have relaxed the assumption to a sign-monotonicity condition: \\(\operatorname{sign}(h_j(D^{\prime}) - h_j(D)) \le 0 \quad \text{for all } j,\\) or \\(\operatorname{sign}(h_j(D^{\prime}) - h_j(D)) \ge 0 \quad \text{for all } j.\\) This relaxed condition is independent of \\(k\\) and holds in several practical settings where the query statistics move in a single direction under a neighbouring change. For example, for \\(D = (7,2,15,6,1,4), D^{\prime} = (7,2,15,3,1,4),\\) the resulting query statistics satisfy a consistent sign pattern. Under this relaxed assumption, Theorem 2 continues to hold, as shown in our updated proof. In summary, we acknowledge that consistency does not hold universally, but the relaxed version remains meaningful in scenarios where monotone changes naturally arise (e.g., rank-based or monotone scoring statistics).
>
> Q2： Lemma 1 looks like a standard post-processing result to me. I am curious if Gumbel DP also satisfies post-processing property. If yes, is it a new property specific to Gumbel DP or property inherited from the general \\(f\\)-DP?
>
> A2： Thank you for the question. Lemma 1 is not a standard post-processing property. Classical post-processing states that \\(g(\mathcal{M}(D))\\) preserves differential privacy for a single query to the database. In contrast, Lemma 1 concerns: (1) \\(k\\) independent Gumbel mechanisms, (2) applied to \\(k\\) separate database queries, (3) followed by the operation \\(g = \min\\). Thus, the input to \\(g\\) comes from multiple queries, not a single one. Moreover, Lemma 1 holds only for the Gumbel mechanism and only for \\(g = \min\\). Therefore, it is not a generic property inherited from $f$-DP, but a structural property specific to Gumbel noise.
>
> Q3： The utility metric shown in 3.2 is not entirely convincing. The authors use noise variance to analyze the privacy-utility performance. However, the private-\\(k\\)  selection problem has a more common and meaningful metric that is the difference between the output statistics and the true minimum value. The absence of analytical results and comparison suing this metric significant weaken the contribution of this work.
>
> A3： Thank you for raising this point. We agree that variance alone is not the ideal utility metric for private top-\\(k\\) selection. Metrics such as the expected gap between the released statistic and the true minimum are often more informative. In this work, we focus on characterizing the privacy properties of the Gumbel mechanism and do not evaluate a concrete applied task, which prevents us from using the more task-specific metric suggested by the reviewer. We appreciate the suggestion and will incorporate such metrics in future application-driven evaluations, where they can be better defined and meaningfully compared.

---

### Official Review · Reviewer_yS3W · 2025-10-30

**Soundness:** 3
**Presentation:** 3
**Contribution:** 1
**Rating:** 2
**Confidence:** 3

**Summary:**

The paper considers a noise additive DP mechanism using the Gumbel distributed noise (called Gumbel mechanism in the paper). This is well known in case of top-k selection, as the arg max of n additive Gumbel noise mechanisms with limited sensitivity deterministic parts is equivalent to the exponential mechanism which is already well convered in the literature (see, e.g., [Dong et al., 2022](http://proceedings.mlr.press/v119/dong20a/dong20a.pdf)). The paper gives analytical expressions for the trade-off functions of the Gumbel mechanisms and their compositions.

**Strengths:**

- Well written, accurate mathematical description of the trade-off functions of the Gumbel noise-additive DP mechanisms and of their compositions.

**Weaknesses:**

- The contribution remains quite thin in my opinion: it is just the analytical characterization of the trade-off functions of the Gumbel noise-additive mechanisms. The second part, i.e., the top-k selection with Gumbel noise is already well known as it is the exponential mechanism.
- No real contribution in the second part (top-k part): the paper states existing results like that of [Dong et al., 2022](http://proceedings.mlr.press/v119/dong20a/dong20a.pdf)
- There is no utility analysis of the Gumbel mechanism. It remains unclear why would one use it.

**Questions:**

The top-k selection is well known and already covered by the existing literature as it is the exponential mechanism. What is really the benefit of using the noise-additive Gumbel mechanism?

---

> ### Author Response · Authors · 2025-11-17
>
> We thank the reviewer for insightful comments. We have carefully considered your comments and responded to the individual concerns.
>
> W1: The contribution remains quite thin in my opinion: it is just the analytical characterization of the trade-off functions of the Gumbel noise-additive mechanisms. The second part, i.e., the top-\\(k\\) selection with Gumbel noise is already well known as it is the exponential mechanism.
>
> A1: We appreciate the reviewer’s concern. Our contribution is not limited to characterizing the trade-off function. Importantly, we also establish a composition theorem for Gumbel additive noise under \\(f\\)-DP. Such results are not automatically inherited from existing \\(f\\)-DP theory and are essential for iterative or multi-step algorithms. Currently, Gaussian mechanisms are the only widely analyzed additive mechanisms in the \\(f\\)-DP framework; our work provides a new, analytically tractable alternative. Regarding the top-\\(k\\) part: although adding Gumbel noise to indices is equivalent to the exponential mechanism, the extended top-\\(k\\) procedure we study differs from the classical formulation. Prior work requires adding fresh Gumbel noise to the associated query values. Our algorithm outputs the original noised query values without injecting a second layer of noise. This results in a non-independent composition that is not covered by standard exponential-mechanism arguments. We show that, due to the structure of Gumbel noise, this extended procedure is still equivalent to an independent composition of two mechanisms—an equivalence not established in previous literature.
>
> W2: No real contribution in the second part (top-$k$ part): the paper states existing results like that of Dong et al., 2022.
>
> A2: While we rely on Dong et al. (2022) for combining exponential mechanisms, this is not the main contribution of Section 4. Our key results are Lemma 1 and Lemma 2 (Page 7), which provide two new composition theorems tailored to the extended top-\\(k\\) setting. These results are not present in prior work and are needed precisely because the extended algorithm involves a non-independent combination of mechanisms. Thus, Section 4 does contribute new theory beyond existing results.
>
> W3: There is no utility analysis of the Gumbel mechanism. It remains unclear why would one use it.
>
> A3: Thank you for raising this point. While a full utility analysis is outside our scope, we provide two concrete motivations for the noise-additive Gumbel mechanism: 1. Reduced noise magnitude: Under general conditions, the variance of Gumbel additive noise is smaller than that of Gaussian noise under \\(f\\)-DP, meaning less distortion for the same privacy level. 2. Avoiding unnecessary quadratic noise in top-\\(k\\): Our extended top-\\(k\\) algorithm requires injecting noise only once, rather than twice as in the classical exponential mechanism formulation. Together, these observations explain why the Gumbel mechanism is attractive for applications that require both privacy guarantees and low distortion.
>
> To summarize the above, we now address your question directly.
>
> Q: The top-\\(k\\) selection is well known and already covered by the existing literature as it is the exponential mechanism. What is really the benefit of using the noise-additive Gumbel mechanism?
>
> A: Thank you for the question. While top-\\(k\\) selection can be implemented using the exponential mechanism, the noise-additive Gumbel mechanism offers advantages not available in the standard formulation.
> First, our extended top-\\(k\\) procedure avoids injecting a second layer of noise into the query values. The classical exponential-mechanism approach requires adding fresh Gumbel noise to scores, whereas our method adds noise only once to the indices and directly reuses the existing noised values. This reduces data distortion and is not addressed in prior work.
> Second, the extended procedure yields a non-independent combination of mechanisms, so it cannot be treated as a standard exponential mechanism. We show that, thanks to the structure of Gumbel noise, it can nevertheless be represented as an equivalent independent composition—an equivalence not established before.
> Third, compared with the Gaussian mechanism under \\(f\\)-DP, the Gumbel mechanism typically introduces smaller noise variance, improving utility at the same privacy level.
>
> In summary, the noise-additive Gumbel mechanism (1) enables an extended top-\\(k\\) selection rule that avoids unnecessary noise injection, and (2) allows new composition results that are not available for general mechanisms.

---

### Official Review · Reviewer_zLEN · 2025-11-01

**Soundness:** 1
**Presentation:** 2
**Contribution:** 1
**Rating:** 0
**Confidence:** 4

**Summary:**

This paper uses Gumbel instead of the Gaussian mechanism to achieve DP by designing a private top-k selection algorithm. The peeling algorithm is considered for analysis where the queries are perturbed by Gumbel mechanism. The analysis is done with f-DP as it has lossless composition property rather than $(\epsilon,\delta)$-DP. Further, conversion between $(\epsilon,\delta)$- DP and f-DP for the Gumbel mechanism is derived. The authors build on the result of [1], which shows that the exponential mechanism is equivalent to the RNM with Gumbel noise

**Strengths:**

Please see the detailed review under questions textbox

**Weaknesses:**

Please see the detailed review under questions textbox

**Questions:**

The paper seems to have a major technical problem namely the fact that it requires the consistency assumption. Note that the consistency assumption (line 293) state that the k query function are consistent if $sign(h_1 (D' ) − h_1 (D)) = · · · = sign(h_k (D' ) − h_k (D))$ where $sign()$ is the sign function. However this would restrict the possible datasets $D'$ itself and won't hold for any arbitrary neighbouring datasets which will violate the very definition of DP. When any datapoint is replaced, even when the ordering doesn't change, the signs will be different when the replaced point happens to be in bottom k: there will be k-1 zeros in the set of sign of differences and one value will be +1 or -1. And when the ordering changes, this set of signs will be even more diverse.
Elaborating with an example, if we consider $k=4$, and neighbouring datasets $D=\{7,2,15,6,1,4\}$ and $D’=\{7,2,15,3,1,4\}$, we have $h_1(D)=1$, $h_2(D)=2$, $h_3(D)=4$, $h_4(D)=6$, and $h_1(D’)=1$, $h_2(D’)=2$, $h_3(D’)=3$, $h_4(D’)=4$. Here, $sign(h_1(D)-h_1(D’))=0$, $sign(h_2(D)-h_2(D’))=0$, $sign(h_3(D)-h_3(D’))=1$, $sign(h_4(D)-h_4(D’))=1$.
Even in this simple example, the consistency assumption doesn’t hold. Similarly for any given dataset, we can find an arbitrary replacement of a datapoint and construct the neighbouring dataset for which consistency doesn’t hold.

Other comments

2) It will be better to state the probability distribution of Gumbel used as Gumbel (minimum) as it is referred to in literature to avoid confusion

3) In Corollary 1 (Page 5), $\epsilon<=0$ is given as the condition in $(\epsilon,\delta)$-DP in order to satisfy $\mu$-GumDP. Isn't $\epsilon$ supposed to be positive according to the definition of $(\epsilon,\delta)$-DP? It appears to be a typographical error.

4) Similarly, in Corollary 2 (Page 6), is $\delta_k(\epsilon)$ guaranteed to take positive values? If not, the domain of operation has to be clearly mentioned.

5) In Appendix A, Step 6, the convexification step that produces the three-segment trade-off $B_{\mu}(\alpha)$ is not clearly explained. Can you clarify how the breakpoints $\alpha_1$ and $\alpha_2$ and the middle section arise from the bi-conjugate construction, i.e., provide a brief derivation or reference for this step (equation 3)?
6) In Appendix D (Page 14), equations (7) and (8) have "n" as the total number of terms in the summation. However, after the transformation of variables as $y_i=\exp(x_i)$, in the subsequent 2 equations, the number of summation terms is written as "k", which should be "n". Please check the consistency of variables being used. Also, will this change the final expressions derived?

---

> ### Author Response · Authors · 2025-11-17
>
> We thank the reviewer for insightful comments. We have carefully considered your comments and responded to the individual concerns. To help you identify the changes in the revised version, we have hightlight all changes in blue.
>
> Q1： The paper seems to have a major technical problem namely the fact that it requires the consistency assumption.
>
> A1: We agree that the original consistency assumption was too strong. Fortunately, it is only used in Theorem 2 (Page 6). As you suggested, we can relax the assumption: the \\(k\\) query functions are consistent if either \\(\\operatorname{sign}(h\_j(D') - h\_j(D)) \\le 0 \quad \\text{for all } j=1,\\dots,k,\\) or \\(\\operatorname{sign}(h\_j(D') - h\_j(D)) \\ge 0 \quad \\text{for all } j=1,\\dots,k.\\)
>
> Under this assumption, the proof of Theorem 2 in Appendix E (Page 15) still holds. In particular:
>
> \\[
> \\begin{aligned}
> T(\\mathcal{M}(D), \\mathcal{M}(D'))
> &= T\\Bigl( \\mathrm{Gumbel}(h\_1(D),\\gamma) \\times \\cdots \\times
> \\mathrm{Gumbel}(h\_k(D),\\gamma),
> \\mathrm{Gumbel}(h\_1(D'),\\gamma) \\times \\cdots \\times
> \\mathrm{Gumbel}(h\_k(D'),\\gamma) \\Bigr) \\\\
> &\\ge T\\Bigl(
> \\mathrm{Gumbel}(0,1)^k,\
> \\mathrm{Gumbel}(\\operatorname{sign}(h\_1(D')-h\_1(D))\\mu,1) \\times \\cdots
> \\Bigr) \\ge B\_\\mu^k .
> \\end{aligned}
> \\]
> Thus, **Theorem 2 remains valid**, this relaxed assumption is satisfied for any arbitrary datapoint replacement. We have updated the consistency assumption in our manuscript on Page 6 of the revision.
>
> Q2: It will be better to state the probability distribution of Gumbel used as Gumbel (minimum) as it is referred to in literature to avoid confusion.
>
> A2: Thank you for pointing this out. Indeed, referring to the Gumbel distribution without specifying "Gumbel (minimum)'' may cause confusion. We have clarified this terminology in our manuscript on Page 4 of the revision.
>
> Q3: In Corollary 1 (Page 5), \\(\\varepsilon<=0\\) is given as the condition in \\((\\varepsilon, \\delta)\\)-DP in order to satisfy \\(\\mu\\)-GumDP. Isn't \\(\\varepsilon\\) supposed to be positive according to the definition of \\((\\varepsilon, \\delta)\\)-DP? It appears to be a typographical error.
>
> A3: Thank you for catching this issue. It was indeed a typographical error. In Corollary 1, the correct condition is \\(\\varepsilon \\ge 0\\),  consistent with the standard definition of \\((\\varepsilon,\\delta)\\)-DP. We have corrected it on Page 5 of the revision.
>
> Q4: Similarly, in Corollary 2 (Page 6), is \\(\\delta_k(\\varepsilon)\\) guaranteed to take positive values? If not, the domain of operation has to be clearly mentioned.
>
> A4: We agree the domain of \\(\\delta_k(\varepsilon)\\) should be stated clearly. The expression arises from the equivalence between \\(f\\)-DP and \\((\\varepsilon,\\delta)\\)-DP in [1], and is given by \\(\\delta_k(\\varepsilon)=1 + {B_{\\mu}^k}^*(-e^{\\varepsilon})\\) where \\(B_{\\mu}^k()\\) is the corresponding trade-off function in \\(f\\)-DP. By construction, this value always lies in \\([0,1].\\)
>
> Q5: In Appendix A, Step 6, the convexification step that produces the three-segment trade-off \\(B_\\mu(\\alpha)\\) is not clearly explained. Can you clarify how the breakpoints \\(\\alpha_1\\) and \\(\\alpha_2\\) and the middle section arise from the bi-conjugate construction, i.e., provide a brief derivation or reference for this step (equation 3)?
>
> A5:  We apologize for not providing a clearer explanation of the convexification procedure. The three-segment form of the trade-off function \\(B_\\mu()\\) arises from minimizing \\(f():=T(\\operatorname{Gum}(0,1),\\operatorname{Gum}(\\mu,1))()\\) and its inverse function \\(f^{-1}()\\) before taking the biconjugate. This procedure mirrors the derivation of Eq.(13) in [1]. Following the same reasoning, the breakpoints \\(\\alpha_1\\) and \\(\\alpha_2\\) naturally emerge as the points where the convex envelope switches between the linear and nonlinear segments. We have included a more explicit explanation on Page  13 of the revision.
>
> Q6: In Appendix \\(D\\) (Page 14), equations (7) and (8) have "\\(n\\)" as the total number of terms in the summation. However, after the transformation of variables as \\(y_i=\\operatorname{exp} \left(x_i\right)\\), in the subsequent 2 equations, the number of summation terms is written as "\\(k\\)", which should be "\\(n\\)". Please check the consistency of variables being used. Also, will this change the final expressions derived?
>
> A6: Thank you for pointing this out. The occurrences of "\\(k\\)" and "\\(n\\)" should indeed be consistent. The correct notation is "\\(k\\)" in all four equations, and the mismatch was a typographical error. Fortunately, this does not affect any of the subsequent derivations or the final results. We have corrected this on Page 14 of the version.
>
> [1] J. Dong, A. Roth, and W. J. Su, “Gaussian differential privacy,” Journal of the Royal Statistical Society Series B: Statistical Methodology, vol. 84, no. 1, pp. 3–37, 2022.

---

> > ### Comment · Reviewer_zLEN · 2025-11-27
> >
> > I thank the authors for their detailed response and effort.
> > However, the new relaxed assumption still appears to hold only for sorting queries and does not extend to arbitrary queries. Any differential privacy accounting mechanism is not useful if it applies only to a narrow class of queries. The authors do not sufficiently justify why an accounting framework restricted to such a limited setting is of practical value. Moreover in my opinion, the improvements claimed for the new Gumbel-DP mechanism are not convincingly supported through theoretical or empirical results.

---

> > > ### Author Response · Authors · 2025-11-28
> > >
> > > We sincerely thank the reviewer for the helpful feedback. We would like to clarify that the relaxed assumption is not limited to sorting queries, but extends to a broader family of coordinate-wise and monotone queries. We also appreciate the reviewer’s point regarding the limited empirical evidence; we acknowledge this shortcoming and will explore more practical applications and additional experiments in future work.

---

### Author Response · Authors · 2025-12-02
**Final Comment to AC**

Dear AC,

We sincerely appreciate the reviewers’ feedback. We addressed the main concerns through theoretical clarifications and revised arguments, documented in the rebuttal (Revision Outline commented) and the revised PDF (highlighted in blue).

- **Reviewer zLEN**  primarily finds that the consistency assumption was too stringent which would restrict the possible datasets
 itself and won't hold for any arbitrary neighbouring datasets which will violate the very definition of DP. In response to this, we have further **relaxed the consistency assumption** and demonstrated that it does not affect subsequent results. Meanwhile, the examples cited by the reviewer satisfy the consistency assumption for relaxation. Moreover, the assumption are applicable to prefix sums queries, a family of “threshold counting” queries, coordinate queries, and so on, demonstrating broad applicability and significant practical value. Second, it points out the lack of clarity in the exposition of the Gumbel distribution, the proof of \\(\delta_k(\varepsilon)\\) in Corollary 2, and the proof of \\(B_{\mu}(\alpha)\\) in Appendix A. We all have provided more detailed explanations.

- **Reviewer yS3W** finds the exponential mechanism has been thoroughly studied in top-\\(k\\) selection problems in Dong et al., 2022. and the utility analysis of the Gumbel mechanism is insufficient. Indeed, the study of exponential mechanisms is well-established, and we have applied existing research findings in our work in Part II. However, we also emphasize that the primary contribution of this paper lies in investigating **additive Gumbel mechanisms** and their combinatorial problems. Concurrently, Part II presents two significant **combinatorial theorems concerning the top-\\(k\\) selection**—areas previously unexplored in prior research on exponential mechanisms. Meanwhile, by comparing the Gaussian mechanism and the Gumbel mechanism in terms of the magnitude of introduced noise variance, we most intuitively demonstrate that the Gumbel mechanism yields superior performance.

- **Reviewer 8q7j**  also identifies issues with the consistency assumption and the utility analysis which have already been addressed in the responses to the first two reviewers. Additionally, reviewer finds Lemma 1 looks like a standard post-processing result. We consider these two concepts distinct. In DP, post-processing refers to applying a function transformation to a query value after DP processing, which still satisfies the DP. In contrast, the approach here involves applying a function transformation to multiple query values processed by DP, and the resulting output may not necessarily satisfy the DP.

- **Reviewer 3Fit**  finds that the CDF of the Gumbel distribution is different from \\(e^{-e^{(x-\\mu) / \\gamma}}\\) which also results in an incorrect proof of Lemma 2. We employ the minimum Gumebl distribution in this work, resulting in a CDF of \\(1-e^{-e^{(x-\\mu) / \\gamma}}\\). To avoid misunderstanding, this has been explicitly stated in the text. Meanwhile, reviewer believes that the invalidity of Inequality  \\(\left|g(\mathcal{D})-g\left(\mathcal{D}^{\prime}\right)\right| \leq \Delta\\)  further indicates that Lemma 2 is problematic. We have supplemented Appendix A with detailed proofs that guarantee the validity of this inequality. Lastly, the Gumbel mechanism is questioned as to whether it is a useful tool, since in Corollary 1, as \\(\mu\\) increases, \\(\delta\\) approaches 1. Through analysis, we find that this phenomenon is consistent with the Gaussian mechanism. And, there exists that both of them have \\(\delta\ll1\\) for sufficiently large \\(\varepsilon\\).

Taken together, the reviewer feedback indicates that our rebuttal and revisions resolved the central technical and conceptual concerns—most notably those regarding the Gumbel (minimum) distribution, the consistency assumption in Definition 9, and the sensitivity \\(\Delta\\) in Lemma 1. Due to the OpenReview incident and the premature closure of the discussion interface, four reviewers were unable to provide follow-up comments or revise their scores after our final clarifications and newly added experiments. We respectfully ask the AC to consider the full written record, including our resolutions to the reviewers’ concerns and the improvements made during the revision process, when making the final decision.

---

### Note · Authors · 2026-01-09

I have read and agree with the venue's withdrawal policy on behalf of myself and my co-authors.